# Comparison of the $U_{37}^{K'}$, LDI, $TEX_{86}^{H}$, and RI-OH temperature proxies in sediments from the northern shelf of the South China Sea

Bingbing Wei[1, 2], Guodong Jia[1], Jens Hefter[2], Manyu Kang[1], Eunmi Park[2, 3, 4], Shizhu Wang[5, 6, 7], and Gesine Mollenhauer[2, 3, 4]

[1]State Key Laboratory of Marine Geology, Tongji University, Shanghai, 200092, China
[2]Alfred Wegener Institute, Helmholtz Center for Polar and Marine Sciences, Bremerhaven, 27570, Germany
[3]Department of Geosciences, University of Bremen, Bremen, 28359, Germany
[4]MARUM, Center for Marine Environmental Sciences, University of Bremen, Bremen, 28359, Germany
[5]First Institute of Oceanography, Ministry of Natural Resources, Qingdao, 266061, China,
[6]Laboratory for Regional Oceanography and Numerical Modeling, Qingdao National Laboratory for Marine Science and Technology, Qingdao, 266071, China.
[7]Key Laboratory of Marine Science and Numerical Modeling, Ministry of Natural Resources, Qingdao, 266061, China.

*Correspondence to*: Guodong Jia (jiagd@tongji.edu.cn), Gesine Mollenhauer (gesine.mollenhauer@awi.de)

**Abstract.** The temperature proxies $U_{37}^{K'}$, LDI, $TEX_{86}^{H}$, and RI-OH are derived from lipid biomarkers, namely long-chain alkenones from coccolithophorids, long-chain diols ascribed tentatively to eustigmatophytes, as well as glycerol dialkyl glycerol tetraethers (GDGTs) and OH-GDGTs produced by Archaea, respectively. The applicability of these proxies in the South China Sea (SCS) has been investigated previously. However, in each study only one or two of the proxies have been compared, and the recently updated calibrations or new calibrating methods such as BASYPAR and BAYSPLINE have not been applied. Here, we investigate four proxies in parallel in a set of surface sediment samples from the northern SCS shelf and relate them to local sea surface temperature (SST), which allows us to compare and assess similarities and differences between them, and also help improve regional multi-proxy seawater temperature reconstructions. Our results indicate that $U_{37}^{K'}$ reflects annual mean SST with a slight bias toward the warm season. Terrestrial inputs appear to have a significant impact on LDI, $TEX_{86}^{H}$, and RI-OH proxies near the coast, leading to colder LDI and $TEX_{86}^{H}$ derived temperatures, but a warmer RI-OH temperature estimate. After excluding samples influenced by terrestrial materials, we find that LDI-derived temperature agrees well with annual SST, while $TEX_{86}^{H}$ and RI-OH derived temperature estimates are close to SSTs in seasons dominated by the East Asian winter monsoon and summer monsoon, respectively. The different seasonal biases of these temperature proxies provide valuable tools to reconstruct regional SSTs under different monsoonal conditions.

## 1 Introduction

Over the past three decades, organic proxies have been successfully applied to reconstruct the Quaternary or even Cenozoic history of sea surface temperature. The two most widely used proxies are the $U_{37}^{K'}$ from alkenones (Brassell et al., 1986) and the $TEX_{86}$ based on archaeal isoprenoid tetraethers (Schouten et al., 2002). More recently, two additional organic

thermometers, the LDI (long chain diol index) and RI-OH (ring index of hydroxylated isoprenoid glycerol dialkyl glycerol tetraethers (OH-GDGTs)), have been proposed by Rampen et al. (2007) and Lü et al. (2015), respectively.

The $U_{37}^{K'}$ proxy is based on the degree of unsaturation of $C_{37}$-alkenones that are synthesized by a very restricted group of haptophyte algae, dominated by the coccolithophores *Emiliania huxleyi* and *Gephyrocapsa oceanica* in typical marine environments (Farrimond et al., 1986; Volkman and L, 1995). Haptophyte algae are light-dependent and live near the sea surface with a competitive advantage in phosphorous-limited environments (Müller et al., 1998; Paasche, 2002). The $TEX_{86}$, as well as its modified versions $TEX_{86}^{H}$ and $TEX_{86}^{L}$ (Kim et al., 2010), is based on the relative abundance of isoprenoid glycerol dialkyl glycerol tetraethers (iGDGTs) containing 1–3 cyclopentane moieties (iGDGT-1, 2, 3, respectively) or 4 cyclopentane moieties with an additional cyclohexane moiety (the crenarchaeol isomer, Cren'). These molecules are produced by marine planktonic Thaumarchaeota (Schouten et al., 2013a). Thaumarchaeota play an important role in pelagic ammonia oxidation in marine environments and tend to maximize in abundance at subsurface depths <200 m (Schouten et al., 2013a). The LDI index is derived from long-chain diols (LCDs), which are likely produced by eustigmatophyte algae (Rampen et al., 2007, 2014a). Comparatively little is known about the biological sources of OH-GDGTs and their influence on the RI-OH proxy. Culture studies suggest that Thaumarchaeota Group 1.1a (e.g., *Nitrosopumilus maritimus*) (Elling et al., 2014, 2015, 2017; Lipp and Hinrichs, 2009; Liu et al., 2012b), SAGMCG-1 (e.g., *Nitrosotalea devanaterra*) (Elling et al., 2017), and a strain of thermophilic euryarchaeota *Methanothermococcus thermolithotrophicus* could synthesize OH-GDGTs (Liu et al., 2012b).

Due to the distinctive ecology of their source organisms (e.g., depth habitat, seasonal blooms), the temperature signals from these biologically derived proxies may differ substantially between each other. For instance, in the basin of the South China Sea (SCS), the $U_{37}^{K'}$ and $TEX_{86}^{H}$ indices likely reflect annual mean temperatures of the mixed layer (≤30 m) (Pelejero and Grimalt, 1997; Jia et al., 2012) and the subsurface (30–125 m) (Jia et al., 2012), respectively. However, in the shallow coastal area of the SCS, the $U_{37}^{K'}$-derived SST estimates are biased toward the spring and summer temperatures, but those based on $TEX_{86}^{H}$ toward the winter temperature, likely due to different blooming times (Zhang et al., 2013). By contrast, in the East China Sea (ECS), both $TEX_{86}^{H}$ and RI-OH signals have been interpreted as reflecting summer temperatures (Lü et al., 2014, 2019; Zhang et al., 2017). Along the Australian southern and eastern coasts, $U_{37}^{K'}$ and LDI provide better estimates of winter temperature at the surface, and $TEX_{86}^{H}$ matches well with annual temperature within 75–100 m (Smith et al., 2013).

The accuracy of organic thermometers can also be affected if the biomarkers they are based on can be synthesized by different (plankton) species. For example, the noncalcifying haptophyte genera *Isochrysis galbana* and *Chrysotila lamellose* also produce alkenones but are predominantly restricted to non-marine and marginal settings. Their alkenone records are distinctly different from those of the open marine species, which is why the $U_{37}^{K'}$ signals in marginal seas are often difficult to interpret (Bijma et al., 2001). GDGTs synthesized by soil archaea and marine Euryarchaeota are likely different in composition from those produced by marine Thaumarchaeota, and contributions of GDGTs to the sediments from these sources could thus introduce biases to $TEX_{86}$ values (Schouten et al., 2013a; Turich et al., 2007; Weijers et al., 2006).

Recently, the LDI proxy was found to depart from an expected calibration because of the input of 1,13 and 1,15-diols from fresh-water eustigmatophyte algae, especially in coastal seas (Balzano et al., 2018; de Bar et al., 2016; He et al. 2020). Similarly, OH-GDGTs may also occur in terrestrial environments including rivers (Chen et al., 2016; Kang et al., 2017), lakes (Liu et al., 2012a), and soils (Kang et al., 2017), which could bias the RI-OH index in marginal seas. Further environmental and physical factors that may bias these proxies include: (1) lateral advection (Benthien and Müller, 2000; Kim et al., 2009); (2) different resistance to degradation (Goni et al., 2001; Kim et al., 2009); and (3) nutrient stress and light limitation (Hurley et al., 2016; Park et al., 2019; Prahl et al., 2003; Versteegh et al., 2001).

Coastal seas are an ideal place for how the organic temperature proxies are influenced by the various confounding factors due to the environmental and ecological seasonality in the transition zone from shallow to deep-sea settings. In this study, we analysed alkenones, LCDs and GDGTs in surface sediments from the northern SCS shelf and continental slope, with a number of samples retrieved from locations shallower than 30 m. All of the proxies $U_{37}^{K'}$, $TEX_{86}^{H}$, LDI, and RI-OH have been previously studied in the northern SCS, a (sub)tropical monsoon climate region (Chen et al., 2018; Ge et al., 2013; Jia et al., 2012, 2017; Lü et al., 2015; Wei et al., 2011; Yang et al., 2018; Zhang et al., 2013; Zhu et al., 2018). However, in these previous studies, only one or two of the proxies were investigated in parallel. Moreover, the updated calibrations or new calibrating methods such as Bayesian calibration models (Tierney and Tingley, 2014, 2018) need to be applied and examined in such a shallow coastal environment, where hydrography and nutrient dynamics are distinctly different from those in the open northern SCS (Wong et al., 2015) and their influences on proxies are still incompletely known. Here, we investigate all four of the above-mentioned temperature proxies in the same surface sediment samples from this area and compare how they correspond to local SSTs. This effort is intended to help improving regional multi-proxy seawater temperature reconstructions, which could be more comprehensive and objective than those based on any single ones (Eglinton and Eglinton, 2008). In addition, such a kind of investigation can shed light on the ecology of the related biomarker producers in this region, which is not entirely understood at present.

## 2 Material and methods

### 2.1 Study area and sample collection

The Pearl River estuary (PRE) and the northern shelf of SCS lie in a (sub)tropical monsoon climate region, with two contrasting monsoon seasons: the East Asian summer monsoon (EASM) and winter monsoon (EAWM). The EASM generally lasts from May to September (Feng et al., 2007; Wang et al., 2004) and the EAWM typically from December to next February (Koseki et al., 2013; Wu, 2016). During the EAWM periods, northeasterly winds are strong, cold, and dry, leading to surface cooling, intensified vertical mixing, and hence an increase of upward nutrient supply to surface waters (Tseng et al., 2005). During the EASM periods, warm southwesterly winds rich in moisture, induce wet conditions, coastal upwelling (Jing et al., 2009), and enhanced freshwater input from the Pearl River, the second largest river in terms of

discharge in China. The outflow of the Pearl River delivers large amounts of nutrients to the coastal northern SCS (Yin et al., 2000).

The northern SCS is a typical oligotrophic sea, characterized by nitrogen-limited conditions like most open oceans (Chen and Chen, 2006), while the Pearl River water is rich in nutrients characterized by a nitrogen/phosphorus (N/P) ratio >100 (Xu et al., 2008; Yin and Harrison, 2008). Thus different nutrient regimes prevail in the northern SCS: primary production is stimulated by discharge of the Pearl River on the inner shelf during the EASM (Chen and Chen, 2006); in the open basin and on the shelf during the EAWM, nutrients are supplied by vertical mixing (Wong et al., 2015).

In the study area, a total of 23 core top sediments (0–1 or 0–2 cm depths) were collected between 2011 and 2017 (Table S1) from the PRE and the northern SCS, from water depths (WD) ranging from 6.5 to 1307 m (Table S1). Most (n =15 of 23) of them were recovered from the inner shelf (WD <50 m), seven from the outer shelf (WD = 50–200 m) and one from the continental slope (LD-21, WD = 1307 m) (Fig. 1a). The samples were collected using a gravity box corer or grab sampler and then stored frozen at −20 °C in the laboratory before treatment.

## 2.2 Lipid extraction and separation

After freeze-drying and homogenizing, about 5 g of sediments were ultrasonically extracted three times with DCM: MeOH (9:1, v/v) for 15 min. Before extraction, known amounts of 2-nonadecanone, androstanol and $C_{46}$ GDGT were added as internal standards. Supernatants of each extraction were obtained by centrifugation. The total lipid extracts were combined and concentrated with rotary evaporation to ~1 mL, and saponified for 2 h at 80 °C with 1 mL of KOH (0.1 M) in MeOH: $H_2O$ (9:1, v/v). Saponification has been suggested as a crucial clean-up procedure for eliminating interferences from co-eluting wax esters during instrumental analysis of alkenones (Villanueva et al., 1997). The sample pre-treatment we used were also used by some of the participants in the interlaboratory comparison of $TEX_{86}$ analytical methods, where extraction procedures were not found to exert significant and systematic effects on $TEX_{86}$ results (Schouten et al., 2013b). The neutral fractions were extracted with $n$-hexane, and were further separated into alkane, alkenone and alcohol sub-fractions (the latter containing diols and GDGTs) by column chromatography on silica gel using $n$-hexane, DCM: $n$-hexane (2:1, v/v) and DCM: MeOH (1:1, v/v), respectively.

## 2.3 Alkenone analysis and $U_{37}^{K'}$ index

Alkenones were analysed using a 7890A gas chromatograph (GC, Agilent Technologies) equipped with a cold on-column injection system, a DB-5MS fused silica capillary column (60 m, ID 250 µm, 0.25 µm film coupled to a 5 m, ID 530 µm deactivated fused silica precolumn) and a flame ionization detector (FID). Helium was used as carrier gas (constant flow, 1.5 mL/min) and the GC oven was heated using the following temperature program: 60 °C for 1 min, 20 °C/min to 150 °C, 6 °C/min to 320 °C and a final hold time of 35 min. Di-unsaturated ($C_{37:2}$) and tri-unsaturated ($C_{37:3}$) alkenones were identified by comparison of the retention times with a reference sample composed of known compounds (Fig. S1a). Peak

areas were determined by integrating the respective peaks, and concentrations were calculated using the response factor of the internal standard 2-nonadecanone.

The $U^{K'}_{37}$ index was calculated using Eq. (1) after Prahl and Wakeham (1987).

$$U^{K'}_{37} = \frac{C_{37:2}}{C_{37:2} + C_{37:3}} \tag{1}$$

SST was estimated using the calibration of Müller et al. (1998) with an uncertainty of 1.5 °C:

$$SST\ (°C) = \frac{U^{K'}_{37} - 0.044}{0.033} \tag{2}$$

In addition, a Bayesian calibration (BAYSPLINE, Tierney and Tingley, 2018) was also applied, as our annual SSTs were above 24 °C. The analytical uncertainty of $U^{K'}_{37}$ index (0.01) was determined from multiple extractions and analyses of a lab-
135 internal reference standard sediment, which was co-analysed with samples for half a year (n = 24).

## 2.4 Long chain diol analysis

One half of each alcohol fraction was silylated with *N, O*-bis(trimethylsilyl)-trifluoroacetamide (BSTFA)/1% trimethylchlorosilane (TMCS) and acetonitrile (30 µL each) and heated at 60 °C for 1 h. Diols were analysed by gas chromatography-mass spectrometry (GC/MS) on an Agilent 6850 GC coupled to an Agilent 5975C MSD operating in
electron impact (EI) mode with an ionization energy of 70 eV. The GC was equipped with a fused silica capillary column (Restek Rxi-1ms, length 30 m; 250 µm ID, film thickness 0.25 µm). Helium was used as carrier gas at a constant flow rate of 1.2 mL/min. Samples (1 µL) were injected in splitless mode in a split/splitless injector (S/SL) held at 280 °C. The GC temperature program was as follows: 60 °C start temperature, held for 3 min, increased to 150 °C at a rate of 20 °C/min, increased further to 320 °C at a rate of 4 °C/min, and finally held at 320 °C for 15 min. The source temperature of the MS
was set to 230 °C and the quadrupole to 150 °C.

For identification of the diols, the MS was operated in single-ion monitoring (SIM) mode with the following *m/z*: 313.3 ($C_{28}$ 1,13-diol, $C_{30}$ 1,15-diol) and 341.3 ($C_{30}$ 1,13-diol, $C_{32}$ 1,15-diol) (Fig. S1b; Versteegh et al., 1997; Rampen et al., 2012). Fractional abundances of the diols were calculated from their integrated peak areas in the respective mass chromatograms.

The LDI was calculated and converted to SST using Eq. (3) and Eq. (4) from Rampen et al. (2012) (calibration error of SST:
2 °C):

$$LDI = \frac{[C_{30}\ 1,15]}{[C_{28}\ 1,13] + [C_{30}\ 1,13] + [C_{30}\ 1,15]} \tag{3}$$

$$SST\ (°C) = \frac{LDI - 0.095}{0.033} \tag{4}$$

A new calibration (Eq. (5), calibration error: 3 °C) from de Bar et al. (2020) is also applied here, as it totally includes 595 global data with an enhanced geographical coverage compared to the original calibration from Rampen et al. (2012):

$$SST\ (°C) = \frac{LDI - 0.1082}{0.0325} \tag{5}$$

The %C$_{32}$ 1,15 index reflecting riverine input was calculated using Eq. (6) given by Lattaud et al. (2017) as follows:

$$\%C_{32}\ 1,15 = \frac{[C_{32}\ 1,15]}{[C_{32}\ 1,15] + [C_{30}\ 1,15] + [C_{28}\ 1,13] + [C_{30}\ 1,13]} \times 100 \tag{6}$$

During the time of analyses, there was no reference sample for diol measurement in our lab, so the analytical uncertainty of LDI is could not be determined.

## 2.5 GDGT analysis and indices (TEX$_{86}$, TEX$_{86}^{H}$, BIT, MI, RI, and RI-OH)

GDGTs were analysed by high performance liquid chromatography (HPLC) coupled via an atmospheric pressure chemical ionization (APCI) interface to a single quadrupole mass spectrometer (MS), with a method slightly modified from Hopmans et al. (2016). Analyses were performed on an Agilent 1200 series HPLC system and an Agilent 6120 MSD. Separation of the individual GDGTs including the 5-/6-methyl isomers of branched-GDGTs was achieved on two UPLC silica columns in series (Waters Acquity BEH HILIC, 2.1×150 mm, 1.7 µm), with a 2.1×5 mm pre-column of the same material maintained at 30 °C. Mobile phase A and B consisted of *n*-hexane: chloroform (99:1, v/v) and *n*-hexane: 2-propanol: chloroform (89:10:1, v/v/v), respectively. After sample injection (20 µL) and 25 min isocratic elution with 18 % mobile phase B, the proportion of B was linearly increased to 50 % within 25 min, and thereafter to 100 % for the next 30 min. After another 5 min and prior to the analysis of the next sample, the column was re-equilibrated with 18 % B for 15 min. The flow rate was 0.22 mL/min and a maximum back pressure of 220 bar was obtained. The total run time was 100 min.

GDGTs were detected using positive ion APCI-MS and selective ion monitoring (SIM) of their (M+H)$^{+}$ ions (Schouten et al., 2007) or abundant ion-source fragmentation products of OH-GDGTs (Liu et al., 2012). APCI spray-chamber conditions were as follows: nebulizer pressure 50 psi, vaporizer temperature 350 °C, N$_2$ drying gas flow 5 L/min and 350 °C, capillary voltage (ion transfer tube) −4 kV and corona current +5 µA. The MS-detector was set for SIM of the following (M+H)$^{+}$ ions: m/z 1302.3 (iGDGT-0), 1300.3 (iGDGT-1+OH-GDGT-0), 1298.3 (iGDGT-2+OH-GDGT-1), 1296.3 (iGDGT-3+OH-GDGT-2), 1292.3 (GDGT-4+4'/Crenarchaeol+isomer), 1050 (GDGT-IIIa/IIIa'), 1048 (GDGT-IIIb/IIIb'), 1046 (GDGT-IIIc/IIIc'), 1036 (GDGT-IIa/IIa'), 1034 (GDGT-IIb/IIb'), 1032 (GDGT-IIc/IIc'), 1022 (GDGT-Ia), 1020 (GDGT-Ib), 1018 (GDGT-Ic), and 744 (C$_{46}$ standard) (Fig. S1c), with a dwell time of 57 ms per ion.

Quantification of the individual GDGTs was achieved by integrating the respective peak areas. Compound contents were calculated using the response factor obtained from the C$_{46}$ standard and by normalizing to the amount of extracted sediment. Due to the lack of appropriate standards, individual relative response factors between the C$_{46}$ standard and the different GDGTs could not be considered, the obtained concentrations should therefore be regarded as being only semi-quantitative.

The TEX$_{86}$ and TEX$_{86}^{H}$ were calculated following Eq. (7) and Eq. (8), respectively. The uncertainties of both TEX$_{86}$ and TEX$_{86}^{H}$ were 0.01, determined from a lab-internal reference standard sediment, which was repeatedly extracted and co-analysed with samples for 3 months (n = 20).

$$\text{TEX}_{86} = \frac{[\text{iGDGT-2}] + [\text{iGDGT-3}] + [\text{Cren'}]}{[\text{iGDGT-1}] + [\text{iGDGT-2}] + [\text{iGDGT-3}] + [\text{Cren'}]} \qquad (7)$$

$$\text{TEX}_{86}^{\text{H}} = \log(\text{TEX}_{86}) \qquad (8)$$

The $\text{TEX}_{86}^{\text{H}}$ values were converted to SSTs using the calibration of Kim et al. (2010) (calibration error: 2.5 °C):

$$\text{SST} = 68.4 \times \text{TEX}_{86}^{\text{H}} + 38.6 \qquad (9)$$

We also examined the Bayesian, spatially-varying calibration (BAYSPAR, Tierney and Tingley, 2014) for the $\text{TEX}_{86}$ as well as a regional winter calibration based on suspended particulate matter (SPM) in the SCS (calibration error of the latter: 1.2 °C, Jia et al., 2017):

$$\text{SST}_{\text{winter}} = 47.18 \times \text{TEX}_{86}^{\text{H}} + 34.44 \qquad (10)$$

The BIT index was calculated according to Hopmans et al. (2004) including 6-methyl brGDGTs (DeJonge et al., 2013):

$$\text{BIT} = \frac{[\text{Ia}] + [\text{IIa}] + [\text{IIIa}] + [\text{IIa'}] + [\text{IIIa'}]}{[\text{Ia}] + [\text{IIa}] + [\text{IIIa}] + [\text{IIa'}] + [\text{IIIa'}] + [\text{Cren}]} \qquad (11)$$

where Ia is the basic tetramethyl brGDGT; IIa and IIIa are 5-methyl brGDGTs; IIa' and IIIa' are 6-methyl brGDGTs (DeJonge et al., 2013).

The methane index (MI) was calculated using the Eq. (12) given by Zhang et al. (2011):

$$\text{MI} = \frac{[\text{iGDGT-1}] + [\text{iGDGT-2}] + [\text{iGDGT-3}]}{[\text{iGDGT-1}] + [\text{iGDGT-2}] + [\text{iGDGT-3}] + [\text{Cren}] + [\text{Cren'}]} \qquad (12)$$

The ring index (RI) was calculated using the Eq. (13) (Zhang et al., 2016):

$$\text{RI} = 0 \times [\text{iGDGT-0}] + 1 \times [\text{iGDGT-1}] + 2 \times [\text{iGDGT-2}] + 3 \times [\text{iGDGT-3}] + 4 \times [\text{Cren}] + 4 \times [\text{Cren'}] \qquad (13)$$

The RI-OH index was calculated using Eq. (14) from Lü et al., (2015), with an uncertainty of 0.01 determined from a lab-internal reference standard sediment, which was repeatedly extracted and co-analysed with samples for 3 months (n = 20).

$$\text{RI-OH} = \frac{[\text{OH-GDGT-1}] + 2 \times [\text{OH-GDGT-2}]}{[\text{OH-GDGT-1}] + [\text{OH-GDGT-2}]} \qquad (14)$$

Lü et al. (2015) presented sedimentary OH-GDGTs data from the China marginal sea (CMS), including some from the northern SCS. In their data, RI-OH correlated best with the summer SST ($R^2 = 0.87$). Besides, a recent observation in the ECS showed that OH-GDGTs abundance in surface water in summer were two times higher than that in winter (Lü et al., 2019), suggesting higher OH-GDGTs production in summer. Thus, the summer calibration (Eq. (15), calibration error: 0.9 °C) from Lü et al. (2015) was applied:

$$\text{SST}_{\text{summer}} = \frac{\text{RI-OH} - 0.005}{0.057} \qquad (15)$$

## 2.6 Climatological mean temperature data and temperature residuals of proxies

The sedimentation rates are not exactly known for each sampling site but sedimentation rates have been reported to vary spatially from 0.2 to 0.6 cm yr$^{-1}$ (Ge et al., 2014; Liu et al., 2014) in the study region. Accordingly, the 0–2 cm surface sediments represent accumulation of more or less a decade. Considering the age uncertainties, we extracted mean annual and monthly SST data for each sampling site, as well as surface salinity in the study region, during an available decadal period (2005–2017) from the NOAA World Ocean Atlas 2018 (WOA18) on a 0.25° grid resolution (https://www.nodc.noaa.gov/OC5/woa18/woa18data.html). Even though a linear trend (0.031 °C yr$^{-1}$) of SST warming has been reported for the SCS (Yu et al., 2019), a different choice of a reference interval would not result in significantly different mean values. The grid resolution of 0.25° in the database is sufficient to define the climatology of the study region, as the distances between 19 out of 23 sampling sites are >0.25° (Fig. 1a). As the regional climate feature are dominated by the seasonally reversing monsoon winds and the transitions between the two contrasting seasons, i.e. from October to November and from March to April, respectively, are relatively short, the SST data were re-analysed and averaged for the two dominant seasons, i.e. EASM (May to September) and EAWM (December to February). Besides, monthly satellite Chlorophyll-a (Chl-*a*) L3 data were obtained from the Moderate Resolution Imaging Spectroradiometer (MODIS) between 2005 and 2017, and average Chl-*a* values in the EASM and EAWM seasons were calculated according to above definition of seasons.

In our study region, SSTs varied spatially within a small range of about 5.5 °C and 3.5 °C during the EAWM and EASM periods, respectively. Together with non-thermal impacts on SST-proxies, such a narrow range usually leads to poor SST-proxy correlations. Thus, we did not use correlation as a criterion to investigate the preferred season of growth of the biomarker producing organisms. Instead, we considered temperature residuals between calculated temperatures from established calibrations and WOA18-derived SSTs, calculated as:

$$\text{Residual (°C)} = [\text{Proxy-derived temperature}] - [\text{WOA18-derived SST}] \tag{16}$$

## 3 Results

### 3.1 Hydrological and Chl-*a* distributions

The annual mean SSTs of sampling sites from WOA18 dataset ranged between 24.2 °C and 27.0 °C (25.6 °C average), and SSTs showed a clear contrast between EAWM (22.4 °C average) and EASM (28.4 °C average) seasons (Fig. 2a, Table 1). During the monsoon transition periods, SSTs were indistinguishable from the annual mean SSTs: 25.3 °C average in March-April and 26.4 °C average in October-November (Table S1). The mean SSTs displayed an increasing trend offshore with the largest difference of ca. 5.5 °C in the EAWM season between inshore and offshore (Fig. 2a).

Surface salinities were generally high at ~34 and uniformly distributed in the study area during the EAWM season, with slightly lower values of ~33.5 along the coastline. While during the EASM season, surface waters freshened due to high

precipitation and elevated freshwater discharge from the Pearl River, leading to a salinity gradient offshore with the lowest values (<32) in the PRE and the highest (~34) to the east of Hainan island (Fig. 3b). Surface Chl-*a* levels were clearly higher on the inner shelf than on the outer shelf during both the EAWM and EASM seasons (Fig. 3c, 3d). Chl-*a* concentration also exhibited a seasonal contrast, which, however, was different between the inner shelf and the outer shelf: higher Chl-*a* occurring in the EASM season on the inner shelf, and in the EAWM season on the outer shelf (Fig. 3c, 3d).

### 3.2 $U_{37}^{K'}$ and alkenone-derived temperatures

The $U_{37}^{K'}$ index ranged between 0.81 and 0.94 (0.91 average, Table 1), corresponding to a temperature range from 23.3 ± 1.5 °C to 27.1 ± 1.5 °C (26.2 °C average, Fig. 2a) based on the linear calibration proposed by Müller et al. (1998), and from 22.6 ± 1.5 °C to 27.6 ± 2.5 °C (26.3 °C average, Fig. 2a) using the non-linear calibration, BAYSPLINE (Tierney and Tingley, 2018). Both the $U_{37}^{K'}$-SST estimates exhibited similar values within 0.7 °C (0.2 °C average; Table S2). Compared with the WOA18-derived SSTs, the average annual residuals of two calibrations were 0.6 ± 0.8 °C and 0.7 ± 1.0 °C, respectively. $U_{37}^{K'}$-SSTs were mostly slightly higher than annual mean SSTs, except two inshore samples (PRE-A8, WD = 17.5 m; MMDB, WD = 26 m) showing slightly lower $U_{37}^{K'}$-SSTs (Fig. 2a).

### 3.3 LCD distribution and LDI-derived temperatures

Of the total 1,13- and 1,15-diols, the $C_{30}$ 1,15-diol was the most abundant homologue (>80 %) at most sites outside the PRE, followed by the $C_{32}$ 1,15-diol (<15 %) and the $C_{28}$ and $C_{30}$ 1,13-diols (<4 %) (Fig. 4a–4d). However, the $C_{32}$ 1,15-diol was more abundant (>41 %) than the $C_{30}$ 1,15-diol (>19 %) in the PRE sediments (Fig. 4c, 4d). The $C_{28}$ and $C_{30}$ 1,13-diols exhibited a similar spatial distribution pattern as the $C_{32}$ 1,15-diol, showing high relative abundances in the PRE and coastal area (Fig. 4a, 4b, 4d).

A Pearson correlation coefficient (PCC) analysis on the fractional abundances of the LCDs was performed using the SPSS software (https://libguides.library.kent.edu/SPSS/PearsonCorr) to examine the relationships between different diols. Fractional abundances of each LCDs from different samples were set as variables, the strength and direction of association that exists between two variables is determined as the PCC, denoted as r. The results showed that $C_{28}$ and $C_{30}$ 1,13-diols and $C_{32}$ 1,15-diol were significantly correlated with each other (r: 0.56–0.83, *p* <0.005, Table 2). In contrast, these three diols were negatively correlated with $C_{30}$ 1,15-diol (r: −0.68 to −0.90, *p* <0.005, Table 2), with the latter exhibiting an opposite distribution pattern and showing an overall increasing trend towards the offshore (Fig. 4c).

The LDI values of surface sediments varied from 0.56 to 0.98 (Table 1), but were ≥0.90 at most sites, corresponding to LDI-derived temperatures (LDI-SST) varying from 14.0 ± 2.0 to 26.9 ± 2.0 °C (Fig. 2b) based on the calibration proposed by Rampen et al. (2012), and from 13.8 ± 3.0 to 26.9 ± 2.0 °C (Fig. 2b) using the recently updated calibration from de Bar et al. (2020). There was no statistically difference (0.03 °C average; Table S3) between two sets of SST estimates. The river input index (%$C_{32}$ 1,15) values ranged from 1.9 % to 66.3 %, showing an overall decreasing trend offshore (Fig. 4d).

### 3.4 Distribution of iGDGTs and $TEX_{86}^{H}$-derived temperatures

The iGDGTs were dominated by crenarchaeol ([Cren], 43.2–65.9 %) and iGDGT-0 ([0], 18.1–37.0 %) (Fig. 5a–5f), with their ratios, i.e. [0]/[Cren], ranging between 0.28 and 0.75 (Fig. 5h). Two samples with relatively high values of [0]/[Cren] were from the PRE (0.75, PRE-A8) and the continental slope (0.69, LD-21) (Fig. 5h). The least abundant iGDGTs is the crenarchaeol isomer ([Cren'], 0.8–5.4 %), showing an overall increasing trend offshore (Fig. 5f). The ratio of iGDGT-0 to crenarchaeol isomer, i.e. [0]/[Cren'], maximized at the river mouth (47.3, PRE-A8) and exhibited a declining trend offshore (Fig. 5i). The ratio of iGDGT-2 versus iGDGT-3, i.e. [2]/[3], ranged from 2.6 to 7.2, showing low values at shelf and coastal sites (WD <200 m, 2.6–3.6) but a high value at the slope site (7.2, LD-21) (Fig. 5i). Similar spatial distribution patterns appeared also for the [2]/[Cren] ratio and the MI value, exhibiting low values of 0.07–0.15 for [2]/[Cren] and 0.16–0.26 for MI at shelf and coastal sites, and slightly higher values of 0.25 for [2]/[Cren] and 0.31 for MI at the slope site (LD-21) (Fig. 5g). In addition, higher BIT values (0.49) were found in the PRE, relative to the inshore area (0.1–0.3, WD <50 m) and the offshore area (<0.1) (Fig. 5h).

$TEX_{86}^{H}$ values varied between −0.33 and −0.18 (Table 1), corresponding to SST values of 16.2 ± 2.6 °C to 26.0 ± 2.6 °C based on the global calibration Eq. (9). We also compared results from different calibrations including BAYSPAR (Tierney and Tingley, 2014) and a local winter calibration Eq. (10), which yielded higher values by 0.9 °C and 1.5 °C than those from the calibration Eq. (9), respectively (Fig. 2c, Table S4). Spatially, consistently low temperature estimates were found on the inner shelf, which were generally colder than the EAWM SSTs (Fig. 2c). The mean residuals relative to EAWM SSTs were −2.0 ± 2.3 °C (using the calibration Eq. (9)), −1.1 ± 1.6 °C (BAYSPAR), and −0.5 ± 1.4 °C (calibration Eq. (10)), respectively (Fig. 2c, Table S4). However, $TEX_{86}^{H}$-SSTs of the offshore samples (E503 and LD-21) were relatively high and similar between the different calibrations, which were ca. 2 °C higher than the EAWM SSTs and ca. 1 °C lower than annual mean SSTs, respectively (Fig. 2c, Table S4).

### 3.5 Distribution of OH-GDGTs and RI-OH-derived temperatures

The OH-GDGTs contributed 1.5–4.1 % to the total GDGT pool (Table S5), consistent with the lower OH-GDGT abundance found in (sub)tropical regions (Huguet et al., 2013). The most abundant OH-GDGT is OH-GDGT-2 ([OH-2], 39.2–67.0 %), with high values at shelf and coastal sites (WD ≤186 m) (Fig. 5l), but a low value at the slope site (LD-21). In contrast, the relative abundance of OH-GDGT-0 ([OH-0]) remained low at shelf and coastal sites, but was elevated at the slope site (Fig. 5j).

The RI-OH values varied from 1.57 to 1.79 (Table 1), which agrees with recently reported data for the same region (1.50–1.75) (Lü et al., 2015; Yang et al., 2018). Summer SST estimates, based on the calibration by Lü et al. (2015) for the CMS, were within a range of 27.5 ± 0.9 to 31.4 ± 0.9 °C (Fig. 2d, Table S5). The residuals of RI-OH-SST relative to EASM SST were mainly between −1.0 ± 0.9 °C and 0.9 ± 0.9 °C, except five samples, three in the PRE (PRE-A8, PRE-Y6, and PRE-

Y11) and other two on the outer shelf (LD-11 and LD-18), were biased toward warm SST with residuals up to 3.2 ± 0.9 °C

(Fig. 2d, Table S5).

## 4 Discussion

### 4.1 Seasonality of the $U_{37}^{K'}$ proxy

Although the relationship between $U_{37}^{K'}$ and SST is robust and well supported by culture studies (Conte et al., 1998; Prahl and Wakeham, 1987; Prahl et al., 1988; Sawada et al., 1996; Volkman et al., 1995), the $U_{37}^{K'}$ response to SST has been found to

310 be attenuated in warm environments (>24 °C), with the slope of the regression decreasing by nearly 50 % as $U_{37}^{K'}$ approaches unity (e.g., Conte et al., 2006, Sonzogni et al., 1997, Tierney and Tingley, 2018). In the northern SCS, annual SSTs are generally >24 °C; however, non-linear calibrations for $U_{37}^{K'}$ have not been applied in previous studies. The BAYSPLINE (Tierney and Tingley, 2018) is the latest non-linear calibration, the application of which in this study showed that it yielded temperatures similar (within 0.7 °C) to the linear calibration by Müller et al. (1998). Considering the errors of the linear

calibration (±1.5 °C) and the BAYSPLINE calibration (up to ±2.5 °C, 1σ), there is no difference between the two sets of SST estimates (Fig. 2a). Also, Pelejero and Grimalt (1997) analysed a series of core-top sediments in the SCS basin and found good linear correlations between $U_{37}^{K'}$ and averaged SSTs of various depths (0, 10, 20, and 30 m) and seasons, indicating that the linear relationship between $U_{37}^{K'}$ and SST is still maintained in such a warm environment. This supports the above finding that there is insignificant difference between SST estimates of linear and non-linear calibrations. Nonetheless,

most $U_{37}^{K'}$-derived temperatures were slightly higher than annual mean SSTs, suggesting a seasonal bias to the EASM season (Fig. 2a), especially for samples recovered from WD ≤100 m. Based on a study of a inshore-offshore transect between 33 m and 102 m WD, Zhang et al. (2013) also proposed $U_{37}^{K'}$-SST to be spring- and summer-biased (April-August) in this region. In surface waters of the SCS outer shelf, the coccolithophore *E. huxleyi*, a major alkenone producer, has been shown to be most abundant in the monsoon transition periods, such as in October ($46 \times 10^3$ cells $L^{-1}$) and March ($19 \times 10^3$ cells $L^{-1}$),

somewhat less abundant in July dominated by EASM ($4 \times 10^3$ cells $L^{-1}$) and least abundant in January dominated by EAWM ($2 \times 10^3$ cells $L^{-1}$) (Chen et al., 2007). The lowest abundance of coccolithophores in winter, when Chl-*a* is elevated (Fig. 3c) due to enhanced mixing, likely results from their competitive disadvantage relative to diatoms (Chen et all., 2007). As the SSTs in the monsoon transition periods are close to the annual mean SSTs, the above seasonal changes in the abundance of *E. huxleyi* support our view that $U_{37}^{K'}$ reflects annual mean SST with a slight bias toward the warm season. Nonetheless, we note

that the bias is not significant.

However, on the SCS shelf, spatial and temporal distributions of alkenone producers have not been carefully investigated, especially on the inner shelf, where high surface Chl-*a* levels occur in the EASM season (Fig. 3d). During this period, surface water salinities are relatively low, mainly due to the discharge of the Pearl River. The river water is enriched in nutrients, the impact of which on primary production, however, is largely limited to the areas within the PRE and along the

coast (Fig. 3d). In addition, the nutrient distribution in the river is characterized by high N:P ratios of up to ~100:1 (Dai et al., 2008; Lu and Gan, 2015; Xu et al., 2008; Zhang et al., 2013). We surmise that such an input of an unbalanced nutrient ratio could stimulate the growth, even though not prominent blooms, of alkenone-producing haptophytes, e.g., *E. huxleyi*, in the oligotrophic shelf waters during the EASM period, since both in situ investigations and experiments have reported that *E. huxleyi* have a competitive advantage over other phytoplankton at high N:P ratios (Riegman et al., 1992, Tyrrell and Taylor, 1996). This phenomenon is likely because the species has a great activity of the enzyme alkaline phosphatase, facilitating assimilation of dissolved organic phosphates (Bijma et al., 2001).

## 4.2 LCDs and LDI-derived temperatures

### 4.2.1 Source of LCDs in the surface sediments

The unusually low LDI-derived SST estimates relative to the WOA18-derived SSTs were observed close to the river mouth and on the inner shelf (Fig. 4e) suggest that LDI may be influenced by terrestrial/freshwater sources other than marine producers. Similar findings were reported from the Iberian margin (de Bar et al., 2016), the Gulf of Lion, the Berau Delta, the Kara Sea (Lattaud et al., 2017) and the East China Sea (He et al., 2020), suggestive of terrestrial influence on LCDs compositions. Culture studies show that marine eustigmatophyte algae mainly produce 1,13 and 1,15-diols (Rampen et al., 2007, 2014a; Volkman et al., 1999). In freshwater environments, eustigmatophyte algae primarily produce $C_{32}$ 1,15-diol, especially in stagnant waters during dry seasons, when rivers have low-stands (Häggi et al., 2019; Lattaud et al., 2017; Rampen et al., 2014b). However, $C_{30}$ 1,15-diol is generally found to be dominant both in the marine water column and sediments and are likely produced by marine eustigmatophyte algae (Balzano et al., 2018).

In this study, the co-occurrence of high abundance of $C_{28}$ and $C_{30}$ 1,13-diols and $C_{32}$ 1,15-diol in the PRE and on the inner shelf rather than in the offshore area (Fig. 4a, 4b, 4d) is consistent with the PCC analysis (Table 2), further suggesting a terrestrial/freshwater source of these diols. Such a spatial distribution pattern becomes more apparent when diol compositions in SPM and sediments are illustrated from the PRE to the offshore (Fig. 4g).  In contrast, the negative correlation of $C_{30}$ 1,15-diol with three other diols could be attributed to their different main sources, i.e. marine vs. terrestrial.

### 4.2.2 Influence of riverine LCDs

It has been pointed out that LCDs delivered by rivers can substantially affect LDI temperature estimates in coastal regions close to river mouths (e.g., Lattaud et al., 2017; He et al., 2020). Lattaud et al. (2017) pointed out that %$C_{32}$ 1,15 in the typical marine sediments generally does not exceed a value of 20 %, which may be used as a cut-off for the reliable reconstruction of LDI-SST, and %$C_{32}$ 1,15 >20 % implies an increased contribution of riverine LCDs. In our samples, LDI-derived temperature estimates from two calibrations were similar to the measured annual SSTs at most sites (Fig. 2b), with 6 exceptions at shallow sites (<26 m) in the PRE and on the inner shelf showing temperature values underestimated by as much as $-11.0 \pm 2.0$ °C (Fig. 2b, 4e). We found that the greater underestimations corresponded to %$C_{32}$ 1,15 values that

are >20 % and 4 times higher than those of the other samples, and the samples with %$C_{32}$ 1,15 <20 % had smaller annual residuals ranging between $-0.2 \pm 2.0$ °C and $1.2 \pm 2.0$ °C (Fig. 4e, Table S3). Besides, the %$C_{32}$ 1,15 values correlated positively ($R^2 = 0.66$, $p$ <0.001) with the BIT index that is often used to indicate terrestrial input in the coastal area (Fig. 4f). Thereby %$C_{32}$ 1,15 is also effective to indicate the river input in this region. After removal of data points (n = 6) with %$C_{32}$ 1,15 >20%, indicating significant influence of riverine LCDs, the LDI-SST of the reduced dataset yields a mean annual residuals of $0.3 \pm 0.4$ °C, much lower than those ($1.3 \pm 3.3$ °C) of the full dataset.

### 4.2.3 Seasonality of LDI index

Our results indicate that LDI-SSTs at sites with minimal river influences may reflect annual SSTs (Fig. 2b, Table S3), suggesting unbiased seasonal production of the source organisms of LDI in this study area. Similar results have been reported by Zhu et al. (2014), who found that LDI-SSTs in downcore sediments match well with the local annual SSTs in the northern SCS. Rampen et al. (2007) found comparable annual flux of 1,15-diols at different stations in the Arabian Sea, and suggested that the biological producers of 1,15-diols do not require a high level of nutrients as needed, e.g., by *Proboscia* diatoms producing 1,14-diols. Thus, LDI may reflect annual SST, with low seasonal abundance variations of marine eustigmatophytes in spite of nutrient variations in an annual cycle on the northern SCS shelf. Nonetheless, since regional annual SSTs are indistinguishable from the monsoon transition periods in spring and/or autumn, we cannot rule out the possibility of prominent occurrences of marine LCD producers during these transition periods.

## 4.3 TEX$_{86}^{H}$ and iGDGT-derived temperature estimates

### 4.3.1 Sources of iGDGTs in the surface sediments

In marine sediments, the iGDGT composition may sometimes be impacted, or even controlled by non-thermal factors, e.g., sources of iGDGTs other than Thaumarchaeota (Zhang et al., 2016). Several indices, e.g., the MI (Zhang et al., 2011), BIT (Hopmans et al., 2004), the [2]/[Cren] ratio (Weijers et al., 2011), and the RI (Zhang et al., 2016) have been developed to assess these impacts. Relatively low MI values (≤0.25) were observed at most sites in our study accompanied by low [2]/[Cren] ratios (0.07–0.15) (Fig. 5g). These values may suggest little input of iGDGTs from archaea involved in methane cycling that are typically characterized by high MI values (>0.3) or substantially elevated [2]/[Cren] ratios (>0.2) (Weijers et al., 2011; Zhang et al., 2016). The exception was the slope sample (LD-21), showing a slightly higher MI value (0.31) and a higher [2]/[Cren] ratio of 0.25, which could suggest some contributions from archaea involved in methane cycling. The constantly low BIT values at most sites are typical for marine sediments with little terrestrial impact. The highest BIT value (0.49) observed in the PRE (sample PRE-A8) (Fig. 5g) is similar to data reported by Zhang et al. (2012). As the BIT index in soils generally tends to be >0.9 (Hopmans et al., 2004), the highest BIT value at the site likely indicates a significant input of soil-derived GDGTs. However, the ability of the BIT index to indicate soil input in this region has been recently discounted by the finding that branched GDGTs may be produced in-situ in aquatic systems (Zhou et al., 2014). Nevertheless,

considering that the sample PRE-A8 is located at the upper river mouth and shows the highest %$C_{32}$ 1,15 values as discussed above, we believe that iGDGTs at this site may be impacted to some extent by terrestrial input.

The [0]/[Cren] ratio was also high at the site PRE-A8. This is likely associated with river input, as the [0]/[Cren] ratio has been found to be high (>2) in soils and river sediments likely due to in-situ methanogenic archaea or imported soil-derived methanogens (Wang et al., 2015; Zhu et al., 2011). Slightly different from other iGDGTs, [Cren'] increased with increasing water depth, with the lowest value of 0.8 % found in the PRE (PRE-A8) and ~1.0 % close to the PRE, while it amounted to 5.4 % at the deepest site (LD-21) (Fig. 5f), in agreement with findings from Jia et al. (2017), who report [Cren'] of >4 % in deep-sea sediments in the SCS. This pattern was unlikely caused by input of soil iGDGTs, as [Cren'] in the soils in the catchments of the Pearl River is ~3 % (Wang et al., 2015). [Cren'] as low as 0.2–0.7 %, with a mean of 0.4 %, was observed in the SPM of the lower Pearl River, which was attributed to the predominance of Euryarchaeota (Wang et al., 2015; Xie et al., 2014). This suggests that iGDGTs close to and within the PRE could also be impacted by the input from aquatic archaea other than Thaumarchaeota.

Several studies suggest that tetraether lipids of Thaumarchaeota dwelling in shallow waters are characterized by [2]/[3] ratios <4 and [Cren'] <4 %, whereas lipids derived from "deep-water" Thaumarchaeota are characterized by higher values for these indices (Jia et al., 2017; Kim et al., 2015, 2016). The difference in iGDGT distributions between the two eco-types of Thaumarchaeota is likely due to the use of different enzymes for iGDGTs synthesis (Kim et al., 2016; Villanueva et al., 2015). Based on these criteria, only one sample, i.e. the slope sample LD-21 with a [2]/[3] ratio of 7.2 and [Cren'] of 5.4 % (Fig. 5f, 5i), likely received some contributions from deep Thaumarchaeota. The sample at the second deepest site, i.e. E503 at 186 m, showed a [2]/[3] ratio of 3.6 and [Cren'] of 4.3 % (Fig. 5f, 5i), which suggests only a small contribution from deep Thaumarchaeota. The occurrence of low [2]/[3] ratios and low [Cren'] fractional abundances for most of our study sites is in agreement with the shallow water depths of these sites, as the depth boundary to separate the deep and shallow Thaumarchaeota, although not exactly determined, is likely 200–300 m (Jia et al., 2017; Kim et al., 2015, 2016).

Theoretically, if planktonic archaea are the dominant GDGT producers, the RI values calculated using fractional abundances of all iGDGTs reflect a response to growth temperatures similar to TEX$_{86}$. This results in a positive correlation between the two indices. Accordingly, Zhang et al. (2016) presented the TEX$_{86}$-RI relationship of the global core top dataset, which they proposed to be used as a criterion to evaluate whether the TEX$_{86}$ value of a given sample is influenced by non-thermal factors. We found that most of our sediment data show a good correlation between TEX$_{86}$ and RI (Fig. 6a); however, they lie outside of the 95 % prediction band using the global TEX$_{86}$-RI relationship (Fig. 6b, Zhang et al., 2016), but, with the exception of two samples, within the 95 % prediction of a "shallow-water" TEX$_{86}$-RI relationship (Fig. 6b, Jia et al., 2017). The two exceptional samples (LD-21 and PRE-A8) are thus likely influenced by other factors than temperature as discussed above. We suggest that this "shallow-water" TEX$_{86}$-RI relationship that is different than that of the global core-top dataset is a robust feature. Our study sites receive predominantly shallow Thaumarchaeota input as demonstrated above, and the shallow Thaumarchaeota likely responds to ambient temperature differently from the deep dwelling communities (Jia et al., 2017; Kim et al., 2015, 2016; Taylor et al., 2013; Villanueva et al., 2015; Zhu et al., 2016). Similarly, the TEX$_{86}$ and RI

values from an incubation study of marine Thaumarchaeota (Schouten et al., 2007) are correlated but lie outside of the 95 % prediction band of the global relationship, likely due to differences in the archaeal community between the incubation experiment and natural marine settings (Zhang et al., 2016). Together, this indicates that $TEX_{86}$ is suitable for temperature estimation in our study area and $TEX_{86}$ in most of our sediments likely indicate regional seawater temperatures.

### 4.3.2 Seasonality of $TEX_{86}^H$ index

Based on the above discussion on iGDGTs indices, only two samples, one in the PRE (PRE-A8) and the other on the slope (LD-21), are markedly different from the remaining samples that appear minimally influenced by soil/freshwater-derived archaea and deep-dwelling Thaumarchaeota or methane-cycling archaea. We therefore exclude these two samples from the following examination of temperature signal recorded by the $TEX_{86}^H$ index.

Our $TEX_{86}^H$-SST estimates were $1.0 \pm 2.6$ to $8.8 \pm 2.6$ °C lower than annual SST using the calibration of Kim et al. (2010), similar to previous studies. The temperature estimates were even lower than the coldest monthly SSTs in the shelf area between 10–100 m WD (Fig. 6c, Table S4). The BAYSPAR estimates yielded slightly higher SSTs, with annual residuals being reduced by ~1.0 °C, however, they are still lower than the coldest monthly SSTs (Fig. 6c, Table S4). When compared with the mean EAWM SSTs, the residuals of both calibrations ranged from $-5.4 \pm 2.6$ °C to $1.9 \pm 2.6$ °C (Kim's calibration) and from $-3.7 \pm 2.3$ °C to $1.6 \pm 1.8$ °C (BAYSPAR), respectively (Fig. 6c, Table S4). As these residuals are not much larger than the calibration error, it may be inferred that $TEX_{86}$ proxy on the northern SCS shelf reflects SST during the coldest season. Similar conclusions have been drawn in several previous studies of $TEX_{86}$ in the northern SCS (Ge et al., 2013; Wei et al., 2011; Zhang et al., 2012; Zhou et al., 2014). Support for this inference comes from a recent observation of iGDGTs abundance in surface waters of the SCS shelf, which in winter were three times higher than in summer (Jia et al., 2017).

Furthermore, we noted that different from the global dataset utilized to establish the $TEX_{86}^H$-SST or $TEX_{86}$-SST, which include a large number of deep-sea sediment samples, our data here were exclusively from shallow sediments receiving iGDGTs predominantly from shallow dwelling Thaumarchaeota. The global calibrations might not be suitable for temperature estimation in our study, as indicated by the different $TEX_{86}$-RI relationship of our data from the global relationship as discussed above (Fig. 6b) and the fact that $TEX_{86}$-derived temperatures are even lower than observed SSTs in the coldest month. Therefore, a local "shallow-water" calibration could be more appropriate for temperature reconstruction. Accordingly, the calibration established from winter SPM (i.e. Eq. (10), Jia et al. 2017) in surface waters of the SCS was applied here. This calibration indeed yielded temperatures closer to the EAWM SSTs (Fig. 2c) with reduced residuals and calibration errors ($-2.8 \pm 1.3$ °C to $1.7 \pm 1.3$ °C, Fig. 6c, Table S4). But it is obvious that some of temperature estimates are still slightly below SSTs in the coldest month. This occurrence has been observed around the PRE and was attributed to the minor contributions of iGDGTs 1 to 4 from MG-II *Euryarchaeota* (Wang et al., 2015, 2017). However, it is still in debate whether MG-II *Euryarchaeota* can produce iGDGTs or not (e.g., Lincoln et al., 2014; Schouten et al., 2014; Besseling et al., 2020; Ma et al., 2020) due mainly to lack of cultured representatives of MG-II *Euryarchaeota* presently.

The relatively closer association of TEX$_{86}$ temperature estimates with EAWM SSTs than EASM and annual SSTs suggests that conditions during the EAWM period may be favourable for the bloom of the autotrophic ammonia oxidizing Thaumarchaeota, the activity of which is enhanced at low light availability and high ammonia concentrations (Horak et al., 2018). At present data on seasonal variations of seawater ammonia in the study region are not available. Water column light levels in the EAWM season are generally low due to the reduced solar irradiation, which may foster a preferential occurrence of Thaumarchaeota during the EAWM season, and hence lead to a bias of TEX$_{86}$ temperatures toward EAWM SSTs.

## 4.4 RI-OH and RI-OH-derived temperatures

### 4.4.1 Source of OH-GDGTs and their influences on RI-OH-SST estimates

A few studies have detected OH-GDGTs in marine, river, lacustrine, and soil environments, indicating ubiquitous and multiple sources of OH-GDGTs (Chen et al., 2016; Huguet et al., 2013; Kang et al., 2017; Liu et al., 2012b; Park et al., 2019; Wang et al., 2012). Kang et al. (2017) noted that [OH-0] (OH-GDGT-0) dominates in marine and estuarine environments (56 ± 10 %), but [OH-2] (OH-GDGT-2) is abundant in lake, river and soil environments, which may lead to overestimated RI-OH-SSTs in case of substantial terrestrial input. Consistently, we found higher RI-OH-SST than EASM SST in the PRE (Fig. 2d, Table S5), where terrestrial input is significant. Besides, at site PRE-A8, its iGDGT composition has been found to be influenced by terrestrial input (see section 4.3.1), which also appears to have an impact on OH-GDGT composition.

In addition, like the Thaumarchaeota, the source organism of OH-GDGTs might also exhibit different thermal responses, namely the OH-GDGTs composition of their membrane lipids, between "shallow-water" and "deep-water" communities. Here, we combined our data with previously published sedimentary OH-GDGTs data in the SCS, with water depths ranging between 3 m and 4405 m (Lü et al., 2015; Yang et al., 2018). We found that [OH-0] is psositively correlated with WD ($R^2$ = 0.66, $p$ <0.001, Fig. 5j), but [OH-2] correlated negatively with WD ($R^2$ = 0.53, $p$ <0.001, Fig. 5l). Meanwhile, except one sample (WD = 41 m), two clusters of samples can be separated based on the [OH-0]/[OH-2] ratio, with the ratio value <0.55 for the shallow-water samples (WD <200 m) and >0.55 for the deep-water samples (WD >200 m) (Fig. 7a). This is surprising because the deep-water sediment samples were collected at warmer, lower lattitudes on the slope and in the basin of the SCS (Yang et al., 2018), which should induce more abundant [OH-2] according to eq. (14). We thus speculate that "shallow-water" and "deep-water" communities have different OH-GDGTs compositions, with more [OH-2] in the deep-water community.

Recently, Yang et al. (2018) found that [OH-2] is positivly correlated with SSTs at SST <25 °C using a modified Bligh/Dyer extraction method, but this relation is inversed at higher SSTs (>25 °C). They therefore proposed a different thermal response of archaeal OH-GDGTs at higher temperatures. However, progressive regression analysis of annual SST with [OH-2], as well as with RI-OH, on our data sequentially removed the outliers that lie outside of the 95 % prediction bands of the respective calibrations and showed that both [OH-2] and RI-OH were positively correlated with SSTs (Fig. 7c, 7d). The

annual SSTs of most (n = 11 of 13) data points laying within 95 % prediction were also above 25 °C. We explain the fact that we reach different conclusions than Yang et al. (2018) by the different water depths at which samples considered in the analyses were recovered. Most (n = 17 of 23) samples of Yang et al. (2018) were located in the deep (WD >971 m) basin of the SCS, and their geographical distribution led to an apparent SST increase with WD (Fig. 7b).

    The [OH-2] and RI-OH of seven samples were identified as outliers in our progressive regression analysis (Fig. 7c, 7d).
Three of them correspond to samples taken in the PRE (PRE-A8, PRE-Y6, and PRE-Y11), and one to the samples from the slope (LD-21), similar to where outliers in the iGDGT distribution were recorded. However, the three other outliers (QD00, LD-11, and LD-18) cannot be explained at present. Unlike iGDGTs, there are no indices developed to assess the impact from non-thermal factors on OH-GDGT distributions.

### 4.4.2 Seasonality of RI-OH index

After excluding the seven outliers identified above, we found that temperature estimates using the summer RI-OH-SST calibration (i.e. (15)) correspond well with EASM SSTs on the shelf of the northern SCS (Fig. 2d), with an average residual of $0.0 \pm 1.1$ °C (Table S5). In comparison, if annual and winter calibrations by Lü et al. (2015) were used, the standard errors of residuals would be 2.3 °C and 3.2 °C, respectively (data not shown here), indicating that the summer calibration provides better estimates. If RI-OH is considered to reflect EASM SSTs rather than annual or EAWM SSTs, which likely indicates
that the source organisms proliferate mainly during the EASM season. Such a conclusion is similar to the observation by Lü et al. (2019), who showed that OH-GDGTs in surface water SPM were more abundant in summer than in winter in the ECS. Comparatively, as discussed above, $TEX_{86}^{H}$ is biased to EAWM SSTs in this region. This may indicate that OH-GDGTs and iGDGTs originate from different organisms. However, the source of OH-GDGTs has not been identified yet, and thus more studies on OH-GDGTs in various regions are needed for a better assessment of the proxy.

### 4.5 Implication for paleoclimatic reconstruction

    After excluding samples with obvious signs of terrigenous supply of the respective lipids, we observed close association between measured annual mean or seasonal SSTs and temperature estimates based on the four proxies discussed here. The relatively poor performance of $TEX_{86}$ in this setting may result from more complicated processes that needs further investigations. Overall, the good agreements between measured SSTs and temperature estimates suggest that resuspension
and lateral transport have only minor impacts on the lipid biomarkers in our study area.

    The reconstruction of EASM and EAWM, being controlled respectively by processes occurring in the tropical Indian-Pacific oceans and in high-latitude Siberia, is a prerequisite for the understanding of paleoclimate change in East Asia. As the strengths of the EAWM and EASM appear to be anticorrelated at least during the most recent geological history (e.g., Yancheva et al., 2007), the possibility of reconstructing seasonal temperatures will greatly advance the understanding of this
system. Our comparison of SST proxies reveals their differential seasonal biases and thus reveals a promising multi-proxy approach to reconstruct EASM and EAWM separately. The coastal and inner shelf of the SCS can provide fine sediment

archives accumulated since the early Holocene (Yim et al., 2006; Ge et al. 2014; Gao et al., 2015), which have been retrieved and studied extensively in recent years including the SST reconstructions mostly using alkenones (Kong et al., 2014, 2017; Lee et al., 2019; Zhang et al., 2019). Further paleo-SST studies using the LDI, $\text{TEX}_{86}^{H}$, and RI-OH are thus expected to reveal

the evolution of this important monsoon system in unprecedented detail.

## 5 Conclusions

Temperature estimates based on the $\text{U}_{37}^{K'}$, LDI, $\text{TEX}_{86}^{H}$, and RI-OH proxies were obtained for surface sediments from the northern South China Sea shelf, including the PRE and the coastal area. For these temperature estimates, the most recent calibrations or newest calibrating methods were considered and – for the two more established proxies $\text{U}_{37}^{K'}$ and $\text{TEX}_{86}^{H}$ –

compared with widely used global calibrations. The temperature estimates were then compared with WOA18-derived annual SSTs, as well as SSTs in the EASM and EAWM seasons. This analysis suggests that $\text{U}_{37}^{K'}$ reflects annual mean SST with a slight bias toward the warm season, when the outer shelf is generally oligotrophic and the inner shelf is influenced by the Pearl River input of nutrients characterized by high N:P ratios. Terrestrial inputs have an appreciable impact, but are limited to waters within and proximal to the PRE, on LDI, $\text{TEX}_{86}^{H}$, and RI-OH proxies, leading to cold-biased (LDI and $\text{TEX}_{86}^{H}$) or

warm-biased (RI-OH) temperatures relative to annual mean SSTs. After excluding from the dataset the samples subject to terrestrial input, the temperature estimates based on these proxies could be ascribed to different seasons, which reflects distinctive ecologies of their source organisms as results of seasonal changes in environmental conditions. LDI-SST matched well with annual SSTs, suggesting that marine eustigmatophyte abundance does not vary strongly with nutrient variation in an annual cycle. For the $\text{TEX}_{86}^{H}$ proxy, a local "shallow-water" calibration based on winter surface water SPM in the SCS

appeared to be more appropriate for temperature reconstruction and reduced residuals relative to SSTs in the EAWM season, although an additional cold bias of temperature estimates still exists. In contrast to $\text{TEX}_{86}^{H}$ indices, RI-OH-based temperature estimates seem to reflect EASM SSTs, hence suggesting a different source organism of OH-GDGTs from that of iGDGTs. As these proxies appear to reflect preferentially different seasons, their combined use has the potential to allow reconstructing seasonal SSTs controlled separately by the EASM and EAWM, which may improve our understanding of the

evolution of the East Asian climate system.

## Acknowledgements

This work is supported by the National Natural Science Foundation of China (grant No. 41676030) and the State Key R&D project (grant No. 2016YFA0601104). BW thanks the China Scholarship Council (201706260033) for the support during his stay at Alfred Wegener Institute (Germany). We acknowledge the captain, crew and scientists who participated in the

expeditions for collecting samples used in this study. Three anonymous reviewers are thanked for their constructive comments that help improve the manuscript.

## Data availability

The data produced in this publication will be available from the PANGAEA database: https://doi.pangaea.de/10.1594/PANGAEA.905187.

## Author contribution

GJ and GM conceived and designed the study. BW and MK collected the samples. BW conducted all the proxy analysis and was aided by JH in the instrument maintenance and data analysis. BW wrote the paper with inputs from GJ, GM, JH, EP, and SW. All the authors reviewed the final manuscript.

## Supplement

There is a supplement related to this article.

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

**Table 1: Sampling sites, SST and proxy values from the PRE and northern SCS shelf (water depth (WD), sea surface temperature (SST) obtained from WOA18, RI-OH, ring index of OH-GDGTs).**

| Sites | Latitude (°N) | Longitude (°E) | WD (m) | $SST_{annual}$ (°C) | $SST_{EAWM}$ (°C) | $SST_{EASM}$ (°C) | $U^{K'}_{37}$ | LDI | $TEX_{86}$ | $TEX^{H}_{86}$ | BIT | RI-OH |
|---|---|---|---|---|---|---|---|---|---|---|---|---|
| PRE-Y6 | 22.3 | 113.8 | 6.5 | 25.2 | 22.0 | 28.1 | 0.91 | 0.69 | 0.53 | -0.28 | 0.28 | 1.69 |
| PRE-Y11 | 22.1 | 113.7 | 8.0 | 25.2 | 22.0 | 28.1 | 0.91 | 0.81 | 0.56 | -0.25 | 0.16 | 1.72 |
| SXCB | 21.5 | 112.7 | 14 | 25.7 | 22.4 | 28.3 | 0.90 | 0.90 | 0.49 | -0.31 | 0.15 | 1.62 |
| GLB | 21.7 | 113.0 | 15 | 25.2 | 21.6 | 28.1 | 0.90 | 0.86 | 0.50 | -0.30 | 0.19 | 1.62 |
| PRE-A8 | 22.7 | 113.7 | 17.5 | 25.2 | 22.0 | 28.1 | 0.81 | 0.56 | 0.60 | -0.22 | 0.49 | 1.79 |
| QD00 | 21.1 | 110.8 | 18 | 24.2 | 20.1 | 27.9 | 0.88 | 0.93 | 0.48 | -0.31 | 0.07 | 1.61 |
| LD-GSD | 22.1 | 113.8 | 21 | 25.2 | 22.0 | 28.1 | 0.91 | n.a. | 0.57 | -0.24 | 0.28 | n.a. |
| WSB | 22.0 | 113.7 | 21 | 25.2 | 21.9 | 28.1 | 0.91 | 0.93 | 0.55 | -0.26 | 0.25 | n.a. |
| MMDB | 21.2 | 111.3 | 26 | 24.9 | 20.9 | 28.3 | 0.85 | 0.64 | 0.47 | -0.33 | 0.10 | 1.59 |
| E700 | 21.5 | 112.5 | 26 | 25.6 | 22.2 | 28.3 | 0.93 | 0.94 | 0.48 | -0.32 | 0.08 | 1.60 |
| YJXB | 21.4 | 111.8 | 27 | 25.7 | 22.4 | 28.3 | 0.89 | 0.94 | 0.55 | -0.26 | 0.22 | n.a. |
| E600 | 21.3 | 111.7 | 29 | 25.0 | 21.1 | 28.3 | 0.93 | 0.95 | 0.49 | -0.31 | 0.05 | 1.60 |
| A9 | 22.0 | 114.0 | 35 | 25.2 | 22.1 | 28.1 | 0.92 | 0.93 | 0.49 | -0.31 | 0.06 | 1.58 |
| E701 | 21.2 | 112.7 | 45 | 25.7 | 22.4 | 28.4 | 0.93 | 0.95 | 0.53 | -0.27 | 0.06 | 1.63 |
| QD04 | 20.4 | 111.1 | 47 | 25.7 | 22.6 | 28.5 | 0.91 | 0.94 | 0.49 | -0.31 | 0.06 | 1.57 |
| SW10 | 22.1 | 115.0 | 58 | 25.6 | 23.0 | 28.2 | 0.93 | 0.97 | 0.54 | -0.27 | 0.04 | 1.66 |
| LD-11 | 20.9 | 114.5 | 86 | 26.2 | 23.5 | 28.4 | 0.94 | 0.97 | 0.60 | -0.22 | 0.02 | 1.73 |
| A6 | 21.3 | 114.7 | 88 | 25.9 | 23.2 | 28.3 | 0.94 | 0.97 | 0.56 | -0.25 | 0.05 | 1.65 |
| LD-18 | 20.6 | 113.8 | 88 | 26.2 | 23.3 | 28.5 | 0.92 | 0.98 | 0.60 | -0.22 | 0.03 | 1.73 |
| QD11a | 20.7 | 113.4 | 90 | 26.1 | 23.0 | 28.5 | 0.92 | 0.95 | 0.51 | -0.29 | 0.06 | 1.61 |
| QD41 | 20.1 | 112.1 | 90 | 26.2 | 23.0 | 28.8 | 0.91 | 0.97 | 0.59 | -0.23 | 0.03 | 1.68 |
| E503 | 19.2 | 112.3 | 186 | 26.8 | 23.9 | 29.2 | 0.94 | 0.98 | 0.65 | -0.19 | 0.03 | 1.72 |
| LD-21 | 19.7 | 114.6 | 1307 | 27.0 | 24.2 | 29.1 | 0.94 | 0.98 | 0.65 | -0.18 | 0.03 | 1.64 |

**Table 2: Pearson correlation coefficient analysis of different diols in surface sediments in this study (** $p < 0.005$**).**

| Pearson correlation coefficient | $C_{28}$ 1,13-diol | $C_{30}$ 1,13-diol | $C_{30}$ 1,15-diol | $C_{32}$ 1,15-diol |
|---|---|---|---|---|
| $C_{28}$ 1,13-diol | 1 | | | |
| $C_{30}$ 1,13-diol | 0.83** | 1 | | |
| $C_{30}$ 1,15-diol | −0.68** | −0.90** | 1 | |
| $C_{32}$ 1,15-diol | 0.56** | 0.78** | −0.90** | 1 |

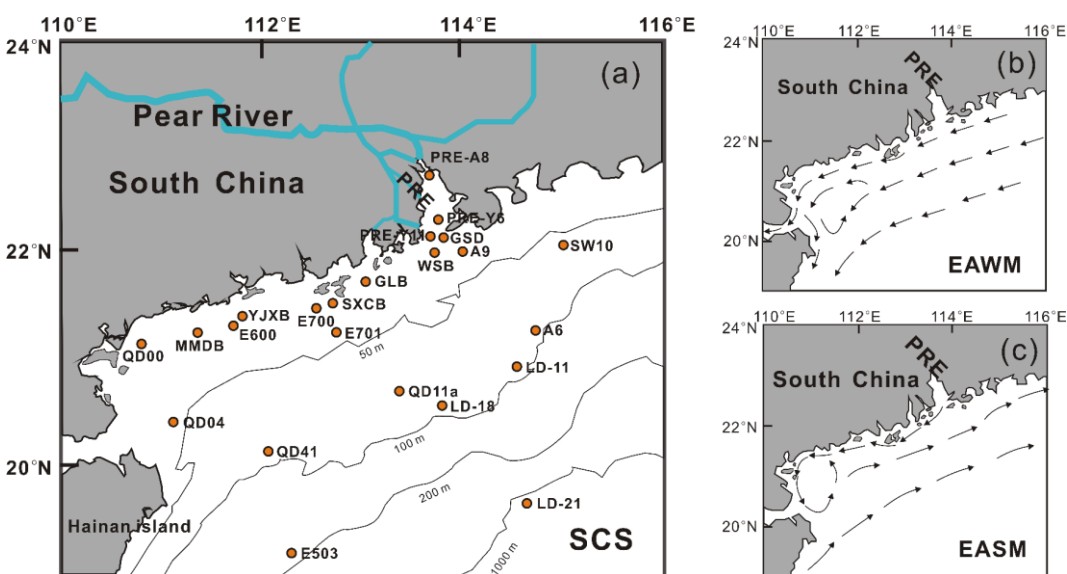

**Figure 1: (a) Sampling sites and patterns of the surface coastal currents in (b) EAWM and (c) EASM seasons (modified from Liu et al. (2014)).**

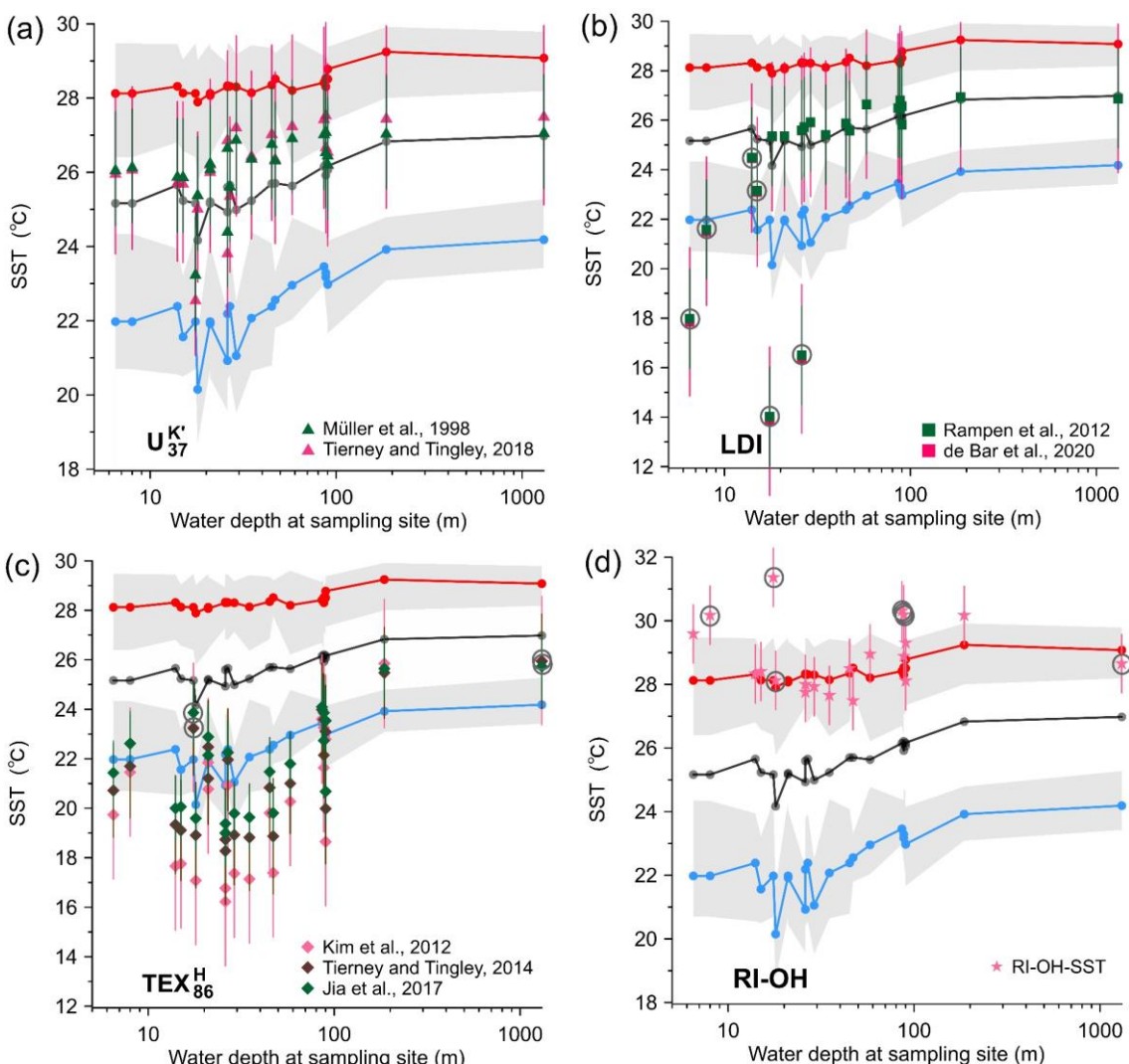

**Figure 2: Changes with water depth of WOA-derived SSTs and proxy-derived SST estimates for: (a) $U_{37}^{K'}$, (b) LDI, (c) $TEX_{86}^{H}$, and (d) RI-OH. Grey, red, and blue dots and lines represent WOA18-drived annual mean, during EASM and EAWM SSTs, respectively. Other symbols represent proxy-derived SST estimates based on (a) $U_{37}^{K'}$ (triangles), (b) LDI (squares), (c) $TEX_{86}^{H}$ (diamonds), and (d) RI-OH (stars). Grey shadings represent SST ranges in each monsoon season. Circled symbols are data likely influenced significantly by non-thermal factors as discussed in the text.**

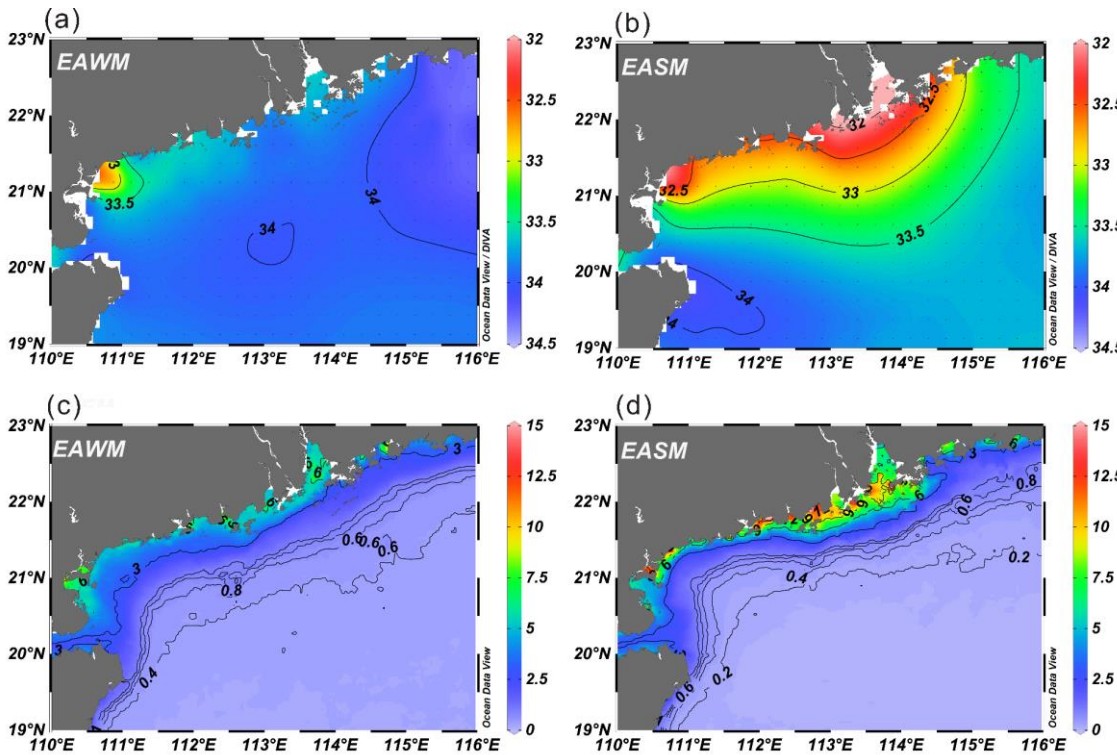

**Figure 3: Spatial distribution of surface salinity in the northern SCS during the (a) EAWM and (b) EASM seasons of 2005–2017 and of surface Chl-*a* concentration (mg m⁻³) in (c) EAWM and (d) EASM seasons of 2005–2017.**

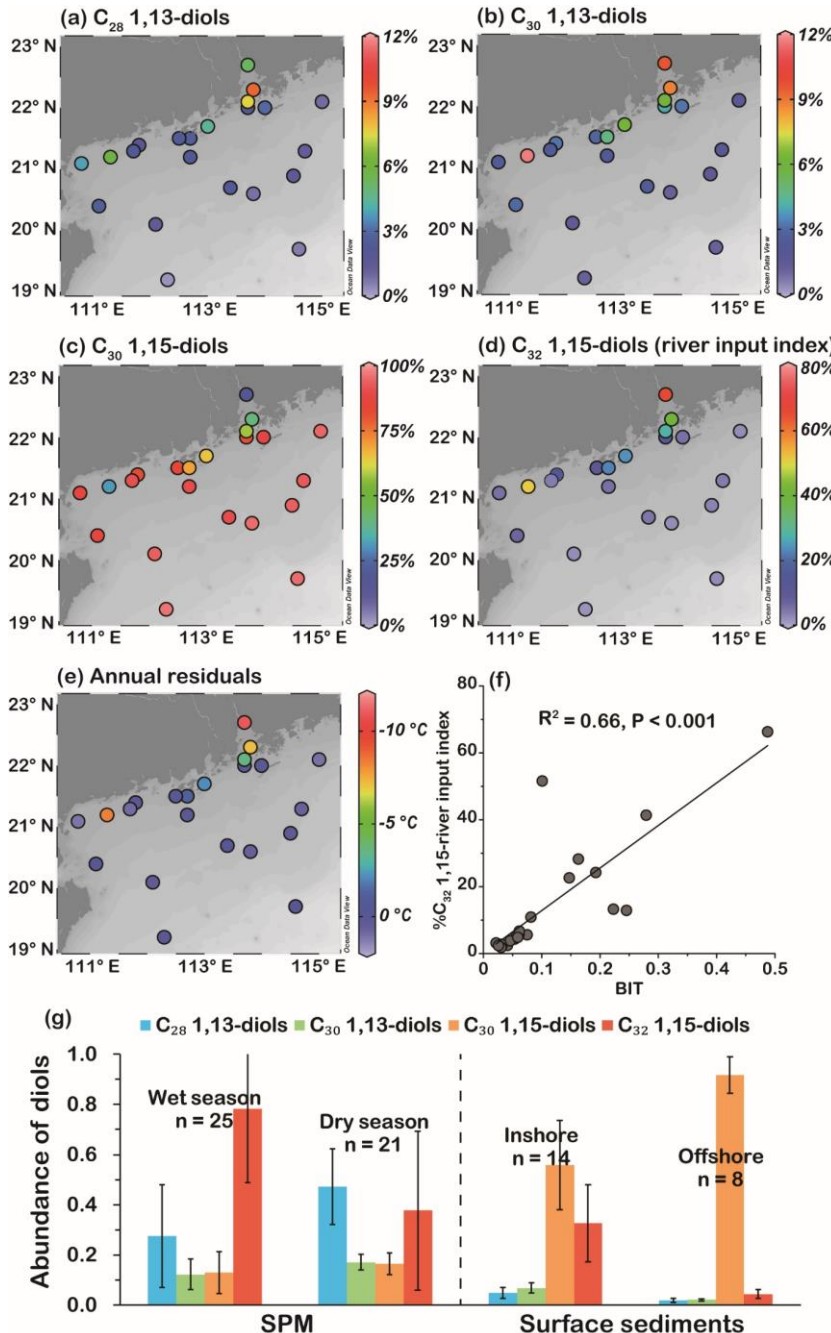

**Figure 4: Spatial distribution of relative abundances of (a) $C_{28}$ 1,13-diol, (b) $C_{30}$ 1,13-diol, (c) $C_{30}$ 1,15-diol, and (d) $C_{32}$ 1,15-diol (river input index), (e) spatial distribution of annual residuals of LDI-derived SSTs, (f) relationship between BIT and %$C_{32}$ 1,15 (river input index), (g) distribution of average fractional abundances of diols in SPM in the PRE (data from Zhu et al. (2018)), inshore and offshore surface sediments (data from this study) (Error bars indicate the standard deviations).**

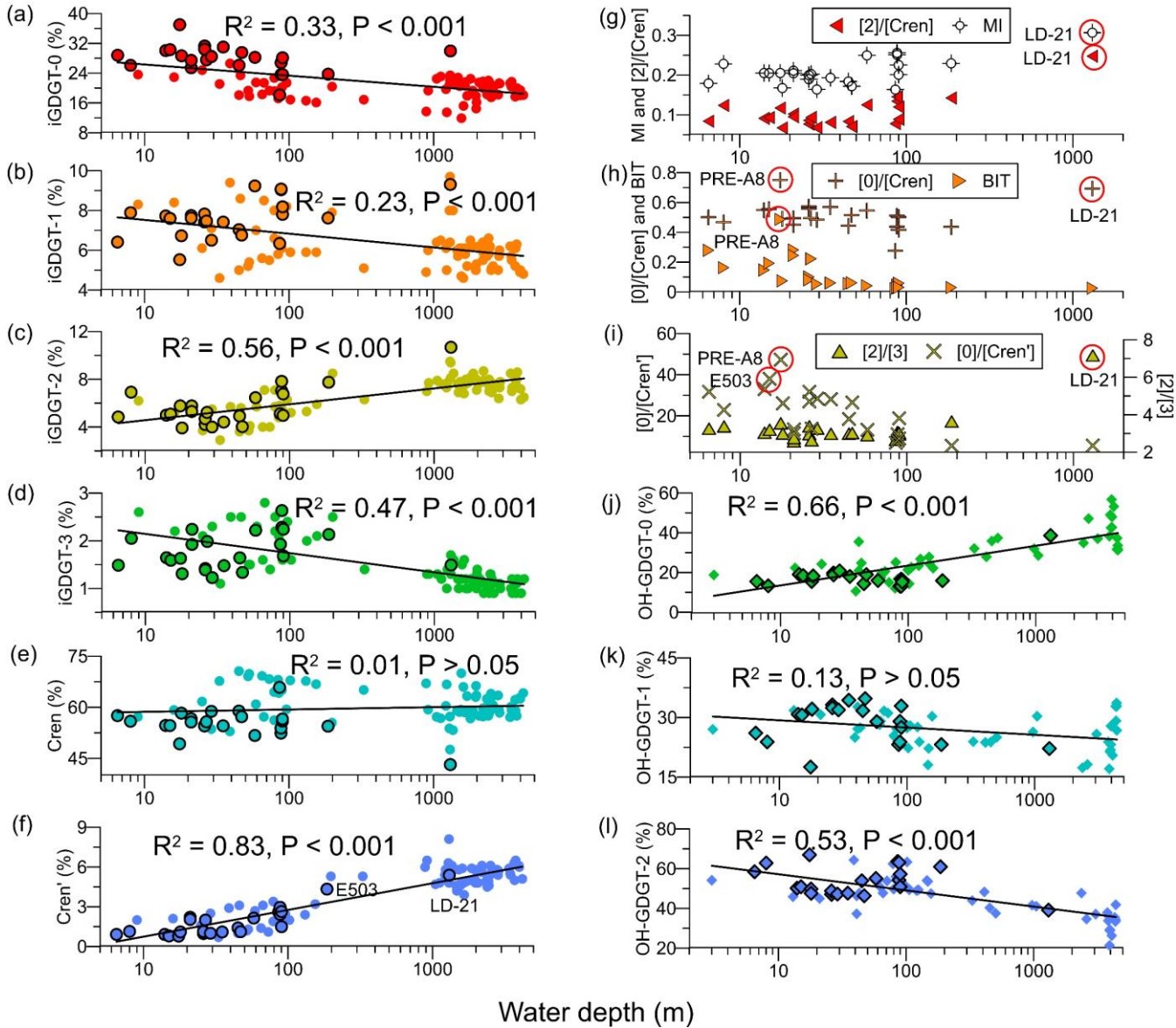

Figure 5: Depth profiles of (a) iGDGT-0, (b) iGDGT-1, (c) iGDGT-2, (d) iGDGT-3, (e) Cren, (f) Cren', (g) [2]/[Cren] ratio and MI values, (h) [0]/[Cren] ratio and BIT values, (i) [2]/[3] and [0]/[Cren']ratios, (j) OH-GDGT-0, (k) OH-GDGT-1, and (l) OH-GDGT-2. Dots with black circles are data from this study and without black circles from Ge et al. (2013), Lü et al. (2015), Jia et al. (2017), Wei et al. (2011), Yang et al. (2018) and Zhou et al. (2014). Black curves are lognormal fits.

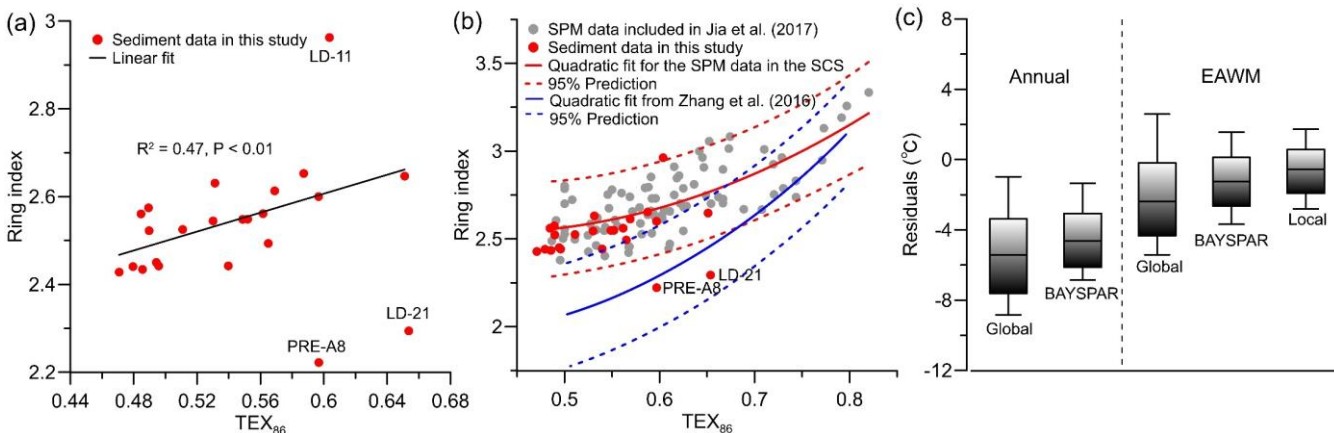

**Figure 6: (a) Relationship of ring index (RI) with TEX₈₆ for surface sediments in this study (black line is the linear fit excluding the data of PRE-A8, LD-11 and LD-21), (b) RI vs. TEX₈₆ scatter plot for surface sediments in this study and surface water SPM from Jia et al. (2017), (c) Box-Whisker plot of annual residuals based on the global calibration (Kim et al., 2010) and BAYSPAR (Tierney and Tingley, 2014), as well as residuals relative to EAWM SSTs based on the global calibration, BAYSPAR, and a local calibration from winter SPM (Jia et al., 2017).**

925

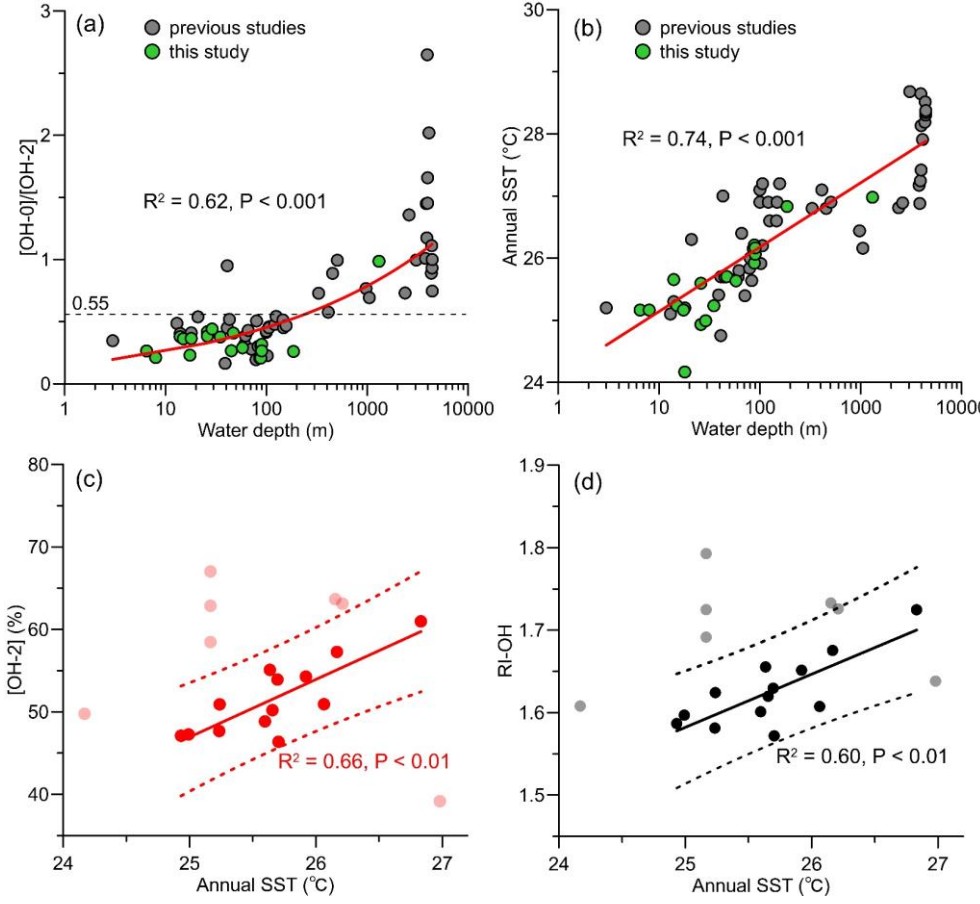

**Figure 7: Changes of (a) [OH-0]/[OH-2] ratio and (b) annual SST with water depth, and relationship of (c) [OH-2] and (d) RI-OH with annual SST. Data from previous studies included in panel (a) and (b) are from Lü et al. (2015) and Yang et al. (2018).**