# Peer review of "Comparison of the $U_{37}^{K'}$ , LDI, TEXH86, and RI-OH temperature proxies in sediments from the northern shelf of the South China Sea"

_Biogeosciences, 2019_

## Referee Comment (RC1) · Anonymous Referee #1 · 16 Oct 2019

**Review: BG-2019-345**

In the manuscript "Comparison of the $U^{K'}_{37}$, LDI, TEX$^H_{86}$ and RI-OH temperature proxies in the northern shelf of the South China Sea," Wei et al. evaluate spatial gradients in four organic paleotemperature proxies across this coastal region. The authors specifically sought to determine the biological sources of the four classes of lipid biomarkers related to the title proxies (alkenones, long-chain diols, isoprenoid GDGTs, and OH-GDGTs, respectively) and any possible seasonal biases in temperatures reconstructed from sedimentary distributions of the lipids. The manuscript offers

insights to the distributions of paleoclimate-relevant lipids in the modern, expanding upon the results of similar studies in the region. The results of this study are particularly relevant for the development of the newer organic paleotemperature proxies based on long-chain diols and hydroxylated GDGTs. Though this work has the potential to inform interpretations of downcore paleotemperature reconstructions from SCS sediments, I find the authors fail to clearly draw this connection themselves. Overall, I feel that the manuscript needs to be significantly revised before it should be considered for publication in Biogeosciences. Specifically, the manuscript could be strengthened with additional statistical analyses that draw more robust connections between environmental variables and lipid distributions as well as with a more in-depth discussion (or at least acknowledgement) of the many factors known to influence lipid production in marine organisms (e.g. growth rates, oxygen concentrations, lateral transport, export efficiency, water column structure) in addition to the few already mentioned in the manuscript.

**General Comments**

The English in this manuscript needs to be improved to increase the readability of the text. The authors are advised to carefully review the entire manuscript for grammatical and syntax errors. Some examples of language that needs improvement are:

Grammatical Errors:

**Line 35:** Need to change "were synthesized" to "are synthesized."

**Line 241:** Authors refer to sediment trap studies (plural) but only include one reference.

Awkward Syntax:

**Line 13-15:** "The applicability of these proxies has been examined in the South China Sea, but most of these studies were focused on a single proxy and hence did not allow for a direct comparison between them."

**Lines 49-50:** "Due to the distinctive ecology of their source organisms, these temperature proxies differ in reflecting water temperatures in terms of, e.g., water depth and seasonality."

**Lines 321-323:** "It should be noted that it remains unclear what causes the different iGDGTs distribution between those two eco-types, and the depth boundary to separate the two, likely 200–300 m, is not exactly determined (Jia et al., 2017; Kim et al., 2015, 2016)."

The discussion section on the seasonal bias of $U_{37}^{K'}$ is weak. Reported measured SSTs approach the upper limits of the Conte et al. (2006) calibration so a nonlinear relationship between $U_{37}^{K'}$ and SST is likely. Linear calibrations, including the one from Conte et al. (2006) used in this study, are hindered by a tendency to underestimate SSTs in warm regions. As such, its possible that the apparent bias towards production of alkenones during cool-seasons inferred from $U_{37}^{K'}$-reconstructed temperatures might be an artifact of this limitation. An example of this in the SCS is illustrated in Tierney and Tingley (2018). Furthermore, support for the authors' hypotheses on the role of nutrients in driving the ecology of alkenone-producers in the SCS is lacking. Why not use nutrient and salinity data provided by the WOA to support your hypotheses on the effect of the PRE on the study region? According to the paper from Chen et al. (2007) cited in this study, alkenone-producer populations in the SCS are more mesotrophic-to-oligotrophic and only outcompete diatoms when nitrate concentrations are relatively depleted. Chen et al. (2007) further state that haptophyte algae populations are sensitive to water column structure, a point that you don't consider in your discussion.

The authors' discussion on iGDGTs is also an incomplete representation of current

knowledge on the TEX$_{86}$ proxy. First, the authors mention that the correlation between TEX$_{86}^H$-reconstructed SSTs and observed SSTs could be improved with a "shallow-water" calibration, yet then proceed to apply the Jia et al. (2017) calibration which is exclusively based on sediments from $> 329$ m water depth. Furthermore, the Jia et al. calibration was calculated against water column temperatures across the upper 30-125 m whereas a number of samples in this study were collected at depths $<30$ m. The authors should consider re-evaluating their data using BAYSPAR (Tierney and Tingley, 2015) *cf.* de Bar et al., 2019 (doi:10.1029/2018PA003453). Second, it is well-known that Thaumarchaeota inhabit a range of depths in the marine water column, but most typically reside at the base of the euphotic zone. A previous study in the SCS identified the depth of maximum Thaumarchaeotal abundance ca. 50 m (Dong et al., 2019), and line 50 of this manuscript states that iGDGTs in the SCS are likely produced between 30 m - 125 m. As many samples in this study lie above these depths, it would be interesting to know how iGDGT concentrations vary across the region and how accumulation rates of sedimentary iGDGTs relate to measured TEX$_{86}$. The authors also reference Zhang et al. (2017), which found that iGDGT distributions in the East China Sea from locations at $<70$ m water depth were significantly impacted by non-temperature influences such as nutrient from upwelling, lateral transport, or resuspension of sedimentary material. These results have significant implications for the interpretation of the data presented here, yet these factors are not thoroughly acknowledged.

The OH-GDGT discussion is complicated by the lack of acknowledgment that the extraction technique employed in this study may have biased the results. Yang et al. (2018) demonstrated that ultrasonic extraction of GDGTs from South China Sediments resulted in significantly lower apparent concentrations of OH-GDGTs relative to samples extracted following a Bligh-Dyer method. In Yang et al., decreased extraction efficiency of OH-GDGTs using an ultrasonic method additionally led to significant biases in SST reconstructions using the RI-OH proxy. You should acknowledge this in

the manuscript in the methods or discussion sections on OH-GDGTs.

For all of the proxy data presented in this manuscript, it would be beneficial to see it placed in the context of other regional studies, similar to Figure 8.

Lastly, the figures do not represent the data well and need to be updated, as do the figure captions which I also found generally uninformative. Figures 2, 3, and 4 are especially difficult to interpret. In Figure 2, the use of the same colors for both the WOA-derived SSTs and the markers related to the proxy data is confusing. This figure could be improved by splitting up the data, for example by having a 4-panel figure with 1-panel per proxy. Figure 3 would be more useful as just a single map showing the spatial trends in the %$C_{32}$1,15 index that is referenced in Line 198 instead of plotting the fractional abundance of each of these 4 diols. For Figure 4, consider averaging the fractional abundances for certain depth classes then plotting the mean for that group in a bar graph similar to your Figure 5. Again, since both the spatial and depth patterns are important in this study, I think it would be best to plot each index in panels 4b and 4d individually as a map similar to your Figure 3.

**Specific comments:**

**Line 39:** Cite Kim et al. (2010) after referring to $TEX_{86}^{H}$ and $TEX_{86}^{L}$.

**Line 40-41:** Crenarchaeol has 4 cyclopentane moieties as well as the cyclohexane ring, so you should rephrase this sentence to something like "...(iGDGTs) containing 0-3 cyclopentane moieties (GDGT-0, 1, 2, 3, respectively) or 4 cyclopentane moieties with an additional cyclohexane moiety (crenarchaeol and its isomer, Cren and Cren', respectively)..."

**Line 41:** Recent studies (Sinninghe Damsté et al., 2018 doi:10.1016/j.orggeochem.2018.06.005; Liu et al., 2018 doi:

10.1016/j.orggeochem.2017.09.009) have determined that the crenarchaeol iso-mer is not actually a regio-isomer, so you should update your text by removing all instances of 'regio.'

**Line 44:** Though there is still a lot to learn about the biological source of LCDs, I don't agree with the statement that the source is "albeit not unambiguously identified yet" as there are several papers that have isolated LCDs in culture studies of diatoms and eustigmatophyte algae, in addition to many phylogenetic studies that link the lipids to source organisms in natural environments.

**Line 45-48** Include references to Elling et al., 2014, 2015 & 2017. Nevertheless, I don't think you represent current knowledge on the source of OH-GDGTs fairly here. See Lipp et al., 2009; Zhu et al., 2016; Sollai et al., 2019.

**Line 167-170:** As nearly half of your samples were collected in years not represented in the WOA13 V2 product, I recommend you update the manuscript with the WOA18 data that was released this summer. Furthermore, given that the studies you contrast your results to later on calculate seasonal averages from different months than those defined in the WOA product, you should use the monthly data from the WOA18 instead of the pre-defined seasonal means to strengthen the comparisons you draw in the discussion section.

**Line 177:** You use the acronym "WD" yet this acronym was never defined previously in the text.

**Line 205/Section 3.4:** You should separate the results related to iGDGTs and those related to OH-GDGTs.

**Line 223-224:** Are these relationships significant?

**Line 233-234:** Why do you refer to the Chinese Marginal Sea here instead of the South China Sea?

**Line 235-236:** It is confusing that you refer to residuals here when the figure you reference (Fig 2) does not show the Proxy − Obs. SST residuals.

**Line 239:** Include references to culture studies that support this statement.

**Line 244-245:** You really should discuss why you think your results indicate that $U_{37}^{K'}$-SST are biased towards spring temperatures as there is nothing in Figure 2 that demonstrates this. You could do a simple linear regression between $U_{37}^{K'}$-derived temperatures and seasonal SSTs and if there is a significant relationship with observation spring SSTs, then your claims are valid.

**Line 270:** Either here or in the methods section you should provide more details on how you conducted this statistical analyses.

**Line 279:** Based on your above discussion on the different sources of these three lipids and the results of the previous studies on LCD distributions in coastal environments that you cite, this supposition about an opposite relationship between $C_{28}1,13$, $C_{30}1,13$, and $C_{32}1,15$ diols and temperature is unlikely and seems unnecessary to include.

**Line 285 - 287:** The statement explaining why a threshold of %$C_{32}1,15$ <20% was used from Lines 288-290 should be moved here.

**Line 305:** Replace "methane-related" with something more accurate such as "...iGDGTs from archaea involved in methane cycling..."

**Line 306:** Please indicate what "substantially elevated" values for the [2]/[Cren] and [0]/Cren indices are as you do for the MI.

**Lines 309-312:** Another paper you cite elsewhere in the text, Zhou et al. (2014), concluded that brGDGTs in the PRE are also likely derived from in situ production in the river rather than solely originating from erosion of catchment soils.

**Line 321-322:** The difference in iGDGT distributions between 'deep' and 'shallow' Thaumarchaeota eco-types is due to the use of different enzymes for iGDGT synthesis (cf. Kim et al., 2016 doi: 10.1016/j.gca.2015.09.010; Villanueva et al., 2014 doi: 10.1111/1462-2920.12508).

**Line 339-343:** In Kim et al.'s 2008 paper, the authors note that the difference in their TEX86 calibration for core top sediments from depths $< 200$ m relative to the calibration for the entire data set is negligible, likely because contribution of iGDGTs from deep-dwelling archaea to the sediment floor is minimal relative to the contributions from shallow-dwelling Thaumarchaeota, a point that has been highlighted in many other studies. As such, I don't believe this is the cause of the mismatch between the $\text{TEX}_{86}^{H}$-reconstructed temperatures and observations you report in your manuscript.

**Line 390:** In Line 315 you draw the opposite conclusion − that your samples are not appreciably impacted by $\text{TEX}_{86}^{H}$?

---

## Referee Comment (RC2) · Anonymous Referee #2 · 26 Nov 2019

This manuscript presents a comparison of four different molecular organic paleotemperature indices (Uk37, TEX86, RI-OH and LDI) from a suite of core-top sediment samples on the continental shelf of the northern South China Sea. They evaluate the spatial gradients across this coastal region and the influence of the Pearl River Estuary on the distribution of these biomarker compounds. Ultimately, they attempt to determine the seasonal preference of the source organisms for each of these proxies by comparing the proxy-derived SST with in situ SST from the WOA gridded data product. This data set is a potentially important contribution to understanding the influence of terrestrial and coastal processes on these marine proxies and could provide constraints on interpretations of downcore proxy-based paleotemperature reconstructions

from the South China Sea. Despite the potential value of this data set, the authors do an incomplete job of analyzing the data and putting it in the context of existing regional data and the body of literature from other near shore oceanographic settings. This manuscript would be suitable for publication in Biogeosciences after substantial revisions that address some of the points below.

One of the most substantive issues in this manuscript is the fact that the conclusions about seasonal bias or weighting of each proxy is based on a simple comparison of residuals between observed seasonal SSTs and proxy-derived SSTs, with an incomplete treatment using the full suite of available calibration equations. There is also only a cursory consideration given to the possible range of depth habitats that each proxy represents in the SCS. Below are some point-by-point comments meant to aid the authors in revising this manuscript with more in-depth analysis that should lead to more robust conclusions.

Comments:

Section 2.1: How as the sedimentation rate determined (1–2 mm/yr, stated in line 170)? The sediment accumulation rate is likely to vary across the offshore transect. Therefore, assuming that all core-top samples represent the mean conditions of a 7-year interval (from 2005–2012) is probably not appropriate. This assumption also does not account for bioturbation, which almost certainly has caused some mixing of material from the past few decades into the upper few centimeters. Making sure that the core-tops in this study are "calibrated" to observational temperatures from the appropriate time interval is especially important given the large SST trends observed over the past decade in the SCS (e.g., Yu, Y., Zhang, HR., Jin, J. et al. Acta Oceanol. Sin. (2019) 38: 106. https://doi.org/10.1007/s13131-019-1416-4).

Uk37-derived temperatures: A number of studies have pointed out the non-linearity of the Uk37-Temperature relationship at SSTs >24–26°C (e.g., Sonzogni et al., 1997, Conte et al., 2006, Tierney and Tingley, 2018). Since SSTs are >24°C for most of

the year at these SCS core sites, it would be worthwhile to calibrate the Uk37 data in this study using BAYSPLINE (Tierney and Tingley, 2018), which accounts for the attenuation of the Uk37 signal at higher SSTs.

Supplemental figure 1. If the Uk37-derived SST ends up being significantly different using alternative uk37-SST equations, it will have implications for the inferred seasonality of the UK37 signal and needs to be addressed more thoroughly in the main body of the paper.

Lines 60–79: There is a substantial body of literature discussing sources of uncertainty and biases in these four biomarkers (especially TEX86 and Uk37) from sediment traps, surface sediments and culture studies. This section should be expanded to include some discussion of those factors. Just a few examples: lateral advection of sediments, light limitation, diagenesis (e.g., preferential degradation of C37:3 in sediments), sensitivity to redox conditions, etc.

Line 167: Why not use the latest WOA18 data?

Line 181–182: Looking at the local hydrographic data, the depth of the mixed layer appears to vary seasonally, with a much deeper winter mixed layer. This should be considered when discussing the potential seasonal and/or depth distribution of the biomarker source organisms.

Overall, there needs to be a much more thorough treatment of uncertainty in the manuscript. There are no error bars or uncertainties shown in any of the figures, nor are they discussed in the results or the supplementary data table. Uncertainty in the various transfer functions used to convert each of these indices to temperature needs to be considered. Analytical uncertainty could be addressed via replicate measurements of samples or standards. The analytical error and calibration error should be propagated and reported when converting proxy index to SST.

Lines 198–204: In order to maintain the organizational flow of the paper, this section

belongs in the Discussion. Also, instead of making the argument that you omit these 6 samples because of their large SST residuals, it makes more sense to omit them because the river input index (%C32 1,15) values are 4x higher than those of the other 19 samples.

Section 4.2.2: Is there any correlation between the BIT and diol river input index in this sample set?

TEX86: There is a large body of literature on the TEX86 proxy that is overlooked in this manuscript. Marine Thaumatchaeota are living throughout the water column in many locations, and it's likely that the TEX86 is integrating the entire water column in these shallow (<200 m) sites.

As with the Uk37-SST equations, I would suggest including BAYSPAR-derived SSTs in the TEX86-SST analysis (Tierney and Tingley, 2015).

Section 4.3.1: The Ring Index (RI), as defined by Zhang et al., 2015, could easily be calculated from the GDGT data used to calculate TEX86 in this study. This is another tool (in addition to the MI, [2]/[cren], and [0]/[cren]) that could be used to screen for non-thermal influences on iGDGT distribution in this sample set.

Figures:

The uncalibrated index values (uk37, LDI, RI-OH and TEX86) are not reported in your figures. I think there should be at least a table that shows the primary data from each of the core-top sites.

Figure 2: This figure is not a very effective way to present these data. I would consider presenting SST maps to show the mean annual, winter and summer SST distribution in the study area (showing spring and autumn is unnecessary in my opinion). Perhaps create separate panels for each of the SST indices. If the authors decide to keep this figure, the lines connecting WOA data points need to be removed, they are distracting. The use of the same colors for the WOA SST data and the proxy-SST data is confusing.

Figure 4: As with Figure 2, panels a and b are not a very informative. Dividing the samples into "inshore and offshore" as is done in Figure 5 and presenting the relative abundances and ratios/indices as bar graphs would be more effective. Again, if the authors chose to keep these panels, refrain from connecting data points with lines.

Figures 3 & 5: The inshore versus offshore comparison of the fractional abundances of diols in figure 5 illustrates the same point as the maps in Figure 3. Therefore, I think Figure 3 is unnecessary. It could be moved to the supplement or removed altogether without losing any information.

Figure 6: I would suggest adding panel d from figure 3 as a second panel in Figure 6 to better illustrate the elevated influence of terrestrially sourced diols in the PRE.

Figure 7: There is no reason to plot the residuals of 3 different calibrations versus the BIT index. If the purpose of the figure is to illustrate that there is no systematic relationship between BIT and SST residuals, then you need only illustrate this using one of the calibrated data sets. If the purpose of the figure is to illustrate the calibration equation that results in the smallest residuals, I would suggest a simpler way of showing the distribution of the data (e.g., box and whisker plots).

Figures 7b & 9: I don't think it's terribly informative to regress any of these indices over a <2 °C temperature gradient (as in the case of summer and autumn SST gradients), however, if you are going to make this plot, why not make a 4-panel figure that does the same for all 4 indices?

Figure 8: The source of the data from a "previous study" needs to be cited in the caption.

Figure 8: In the caption, the authors need to clearly state what "annual residuals" are. Also, "fitting lines" is not a mathematical term. Are these ordinary least squares regression lines? Something else?

Minor Comments:

Lines 49–50: This sentence is awkward to read. I would suggest changing to something like "Due to the distinctive ecology of their source organisms (e.g., depth habitat and seasonal preference), coeval temperature records from each of these proxies may differ substantially"

Line 51: Remove the "however" from this sentence.

Lines 57–59: Remove "while" from the beginning of the sentence. These two sentences seem to contradict each other. Also, what is the southeast Australian Ocean?

Line 60: This sentence doesn't make sense: "The accuracy of organic thermometers is also prone to be impaired by a low specificity of related biomarkers."

Line 83: Should be northeasterly and southwesterly winds

Lines 171–172: I believe WOA13 defines the summer and autumn as Jul–Sept and Oct–Dec, respectively (not Jul–Aug and Sept–Dec as stated here). Line 199: Instead of stating "three samples with low LDI values", it would be more descriptive to state, "three samples with LDI values lower than predicted from local SST".

Line 244: Explain what is meant by "complex sedimentation processes". If the authors are talking about lateral transport of the fine sediment fraction, or diagenetic alteration of the signal, this could be expanded on significantly here.

Lines 257–258: This sentence is confusing and should be rewritten for clarity.

Line 279: It is unclear what is meant by "an opposite response to ambient temperature of C30 1,15-diol to C28 and C30 1,13-diols".

Line 305: Change "methane-related" to methanotophic archaea

---

## Author Comment (AC1) · 14 Dec 2019

Dear reviewer, Thank you for your helpful comments. We have addressed your comments and make changes accordingly. Please find related contents in the document file with changes marked.

General comments: The English in this manuscript needs to be improved to increase the readability of the text. The authors are advised to carefully review the entire manuscript for grammatical and syntax errors. Some examples of language that needs improvement are:

[Figure]

Response: Thanks for the comments. In the revised version, we have checked English expression both in grammatical and syntax.

Line 35: Need to change "were synthesized" to "are synthesized."

Response: Done. Pls see line 36.

Line 241: Authors refer to sediment trap studies (plural) but only include one reference.

Response: In fact there are numerous sediment trap studies, so we change "(Rosell-Melé and Prahl, 2013)" to "(e.g., Rosell-Melé and Prahl, 2013 and references therein)". Pls see lines 265–266.

Awkward Syntax: Line 13-15: "The applicability of these proxies has been examined in the South China Sea, but most of these studies were focused on a single proxy and hence did not allow for a direct comparison between them."

Response: Sentence has been rephrased as "The applicability of these proxies has been examined in the South China Sea (SCS), but only one or two of them were studied in each work. Thereby, it is difficult to make a direct comparison between these proxies in this region." Pls see lines 13–15.

Lines 49-50: "Due to the distinctive ecology of their source organisms, these temperature proxies differ in reflecting water temperatures in terms of, e.g., water depth and seasonality."

Response: Sentence has been rephrased as "Due to the distinctive ecology of their source organisms (e.g., depth habitat and seasonal bloom), the temperature signals from these biologically derived proxies may differ substantially between each other." Pls see lines 51–52.

Lines 321-323: "It should be noted that it remains unclear what causes the different iGDGTs distribution between those two eco-types, and the depth boundary to separate the two, likely 200–300 m, is not exactly determined (Jia et al., 2017; Kim et al., 2015,

2016)."

Response: Sentence has been rephrased as "The difference in iGDGTs distributions between those two eco-types of Thaumarchaeota is due to the use of different enzymes for iGDGTs synthesis (Kim et al., 2016; Villanueva et al., 2015)." (Pls see line 360-361) and "The occurrence of low [2]/[3] ratios and low [Cren'] fractional abundances for most of our study sites is in agreement with the shallow water depths of these sites, as the depth boundary to separate the deep and shallow Thaumarchaeota, although not exactly determined, is likely 200–300 m (Jia et al., 2017; Kim et al., 2015, 2016)." (Pls see lines 365–367).

The discussion section on the seasonal bias of U37K' is weak. Reported measured SSTs approach the upper limits of the Conte et al. (2006) calibration so a nonlinear relationship between U37K' and SST is likely. Linear calibrations, including the one from Conte et al. (2006) used in this study, are hindered by a tendency to underestimate SSTs in warm regions. As such, it's possible that the apparent bias towards production of alkenones during cool-seasons inferred from U37K'-reconstructed temperatures might be an artifact of this limitation. An example of this in the SCS is illustrated in Tierney and Tingley (2018). Furthermore, support for the authors' hypotheses on the role of nutrients in driving the ecology of alkenone-producers in the SCS is lacking. Why not use nutrient and salinity data provided by the WOA to support your hypotheses on the effect of the PRE on the study region? According to the paper from Chen et al. (2007) cited in this study, alkenone-producer populations in the SCS are more mesotrophic-to- oligotrophic and only outcompete diatoms when nitrate concentrations are relatively depleted. Chen et al. (2007) further state that haptophyte algae populations are sensitive to water column structure, a point that you don't consider in your discussion.

Response: Thanks for the comments. (1) We have checked the difference between linear and nonlinear calibrations, especially the BAYSPLINE (Tierney and Tingley, 2018) SST estimate. Indeed, BAYSPLINE SST estimate yields slightly higher temperature values by ∼0.5 °C in average than the linear calibration. But this is not against our

conclusion of spring bias of alkenone temperature but reinforces it. Pls see related statements in lines 218–220 and 274–277. (2) As far as we know, the ecology of alkenone-producers in the SCS shelf is largely unknown except the paper of Chen et al. (2007). Even in that paper, the focus is on the oligotrophic basin and the relatively nutrient replete shelf is only marginally investigated. Their data showed that on the SCS shelf, coccolithophores, especially the alkenone producer E. huxleyi, were lowest during winter, when nutrients are higher, but were abundant during spring, summer and autumn. Our conclusion is basically consistent with their data. (3) The use of the water column structure, nutrient and salinity data in phytoplankton ecology studies is really a common practice. However, our study sites are mostly shallow with water depths of most of them <50 m, where water stratification does not extensively occur and WOA data fail to provide high-resolution data. So we didn't rely our interpretation on those information.

The authors' discussion on iGDGTs is also an incomplete representation of current knowledge on the TEX86 proxy. First, the authors mention that the correlation between TEX86H-reconstructed SSTs and observed SSTs could be improved with a "shallow water" calibration, yet then proceed to apply the Jia et al. (2017) calibration which is exclusively based on sediments from >329 m water depth. Furthermore, the Jia et al. calibration was calculated against water column temperatures across the upper 30-125 m whereas a number of samples in this study were collected at depths <30 m. The authors should consider re-evaluating their data using BAYSPAR (Tierney and Tingley, 2015) cf. de Bar et al., 2019 (doi:10.1029/2018PA003453).

Response: (1) The "shallow-water" calibration from Jia et al. (2017) is based on the surface SPM from 5 m water depth, thereby is really a "shallow-water calibration", but not the "sediments from >329 m water depth" as you said. We thought here you are referring to the work of Jia et al. (2012), in which they pointed out that in the SCS basin sedimentary TEX86 reflects best the temperature at 30–125 water depth. We rephrase related contents in lines 398–403. (2) We re-evaluated our data using BAYSPAR according to the comment, and did not find substantial differences. Comparatively, the local "shallow-water" calibration yielded smallest temperature residuals. Pls see our discussion in lines 392–393 and 398–403.

Second, it is well known that Thaumarchaeota inhabit a range of depths in the marine water column, but most typically reside at the base of the euphotic zone. A previous study in the SCS identified the depth of maximum Thaumarchaeotal abundance ca. 50 m (Dong et al., 2019), and line 50 of this manuscript states that iGDGTs in the SCS are likely produced between 30 m-125 m. As many samples in this study lie above these depths, it would be interesting to know how iGDGT concentrations vary across the region and how accumulation rates of sedimentary iGDGTs relate to measured TEX86.

Response: The line 50 is saying about the scenario happening in the basin of the SCS, where water depth is >300 m (Jia et al., 2012) and the finding of depth of maximum Thaumarchaeotal abundance ca. 50 m is based on iGDGT concentration in seawater SPM (Dong et al., 2019). It is really interesting to know how iGDGTs, and hence Thaumarchaeotal abundance, change with depth on the shallow shelf. We think this question could be answered by means of iGDGTs measurement in particulate matter in the water; but due to that the water depths are quite shallow (mostly <50 m) and the water are well mixed in this study region, we do not expect iGDGT concentration would change substantially with depth. We also don't believe iGDGTs concentration in sediment can do such a work, because iGDGT concentration in sediment is not only determined by Thaumarchaeotal abundance, but also controlled by bulk accumulation rate that is variable among sites. The bulk accumulation rate for each site is unknown in this study, so we are unable to relate accumulation rates of sedimentary iGDGTs with measured TEX86.

The authors also reference Zhang et al. (2017), which found that iGDGT distributions in the East China Sea from locations at <70 m water depth were significantly impacted by non-temperature influences such as nutrient from upwelling, lateral transport, or resuspension of sedimentary material. These results have significant implications for the interpretation of the data presented here, yet these factors are not thoroughly acknowledged.

Response: Lateral transport and resuspension could exert some impacts on the TEX86 proxy in shallow dynamic environment. This factor may be studied through, e.g., a comprehensive comparison between SPM and sedimentary data as did by Zhang et al. (2017). In this work, only sedimentary data were available and hence lateral transport and resuspension can only tentatively acknowledged. Pls see lines 378-382

The OH-GDGT discussion is complicated by the lack of acknowledgment that the extraction technique employed in this study may have biased the results. Yang et al. (2018) demonstrated that ultrasonic extraction of GDGTs from South China Sediments resulted in significantly lower apparent concentrations of OH-GDGTs relative to samples extracted following a Bligh-Dyer method. In Yang et al., decreased extraction efficiency of OH-GDGTs using an ultrasonic method additionally led to significant biases in SST reconstructions using the RI-OH proxy. You should acknowledge this in the manuscript in the methods or discussion sections on OH-GDGTs.

Response: In the revision we mentioned the method of Yang et al. (2018) (Lines 412–417), but we did not say more about it because it beyond this work. However, we expanded our discussion including their data and findings. Pls see related discussion in lines 418–430.

For all of the proxy data presented in this manuscript, it would be beneficial to see it placed in the context of other regional studies, similar to Figure 8. Lastly, the figures do not represent the data well and need to be updated, as do the figure captions which I also found generally uninformative. Figures 2, 3, and 4 are especially difficult to interpret.

Response: We updated figures and figure captions.
In Figure 2, the use of the same colors for both the WOA-derived SSTs and the markers related to the proxy data is confusing. This figure could be improved by splitting up the data, for example by having a 4-panel figure with 1-panel per proxy.

Response: Following your suggestions, in the revision, we made a 4-panel figure with 1-panel per proxy, and used a different color to show proxy-derived data. Pls see related changes in Fig. 2.

Figure 3 would be more useful as just a single map showing the spatial trends in the %C32 1,15 index that is referenced in Line 198 instead of plotting the fractional abundance of each of these 4 diols.

Response: We prefer to keep maps of fractional abundance of each of these 4 diols, because it is obvious to find that some C28 and C30 1,13-diols come from the discharge of the Pearl River. Together with its high positive correlation with C32 1,15-diols, these could explain the unusual low LDI values in the inshore areas. Besides, we also add a map exhibiting annual residuals in study area to better show the relation between annual residuals and fractional abundance of C28 and C30 1,13-diols and C32 1,15-diols. Pls see Fig. 4.

For Figure 4, consider averaging the fractional abundances for certain depth classes then plotting the mean for that group in a bar graph similar to your Figure 5.

Response: Thanks for comments. This is a good suggestion to make the distribution pattern of the individual iGDGTs more clear, but it may be hard to select appropriate depths. For the same goal, we split Fig. 4a into 6 panels and include data from other regional studies. Pls see related changes in Fig. 5a–5f.

Again, since both the spatial and depth patterns are important in this study, I think it would be best to plot each index in panels 4b and 4d individually as a map similar to your Figure 3.

Response: To be honest, for each index, our data only varied in a small range in the

study region, with except of two samples (PRE-A8 and LD-21), which were marked in Fig. 5g–5i.

Specific comments: Line 39: Cite Kim et al. (2010) after referring to TEX86H and TEX86L.

Response: Done. Pls see line 40.

Line 40-41: Crenarchaeol has 4 cyclopentane moieties as well as the cyclohexane ring, so you should rephrase this sentence to something like "...(iGDGTs) containing 0-3 cyclopentane moieties (GDGT-0, 1, 2, 3, respectively) or 4 cyclopentane moieties with an additional cyclohexane moiety (crenarchaeol and its isomer, Cren and Cren', respectively)..."

Response: Rephrased. Pls see lines 41–42.

Line 41: Recent studies (Sinninghe Damsté et al., 2018 doi:10.1016/j.orggeochem.2018.06.005; Liu et al., 2018 doi:10.1016/j.orggeochem.2017.09.009) have determined that the crenarchaeol isomer is not actually a regio-isomer, so you should update your text by removing all instances of "regio.

Response: "regio-" was deleted in the revision.

Line 44: Though there is still a lot to learn about the biological source of LCDs, I don't agree with the statement that the source is "albeit not unambiguously identified yet" as there are several papers that have isolated LCDs in culture studies of diatoms and eustigmatophyte algae, in addition to many phylogenetic studies that link the lipids to source organisms in natural environments.

Response: We change "not unambiguously identified" to "not fully clear". Pls see line 45.

Line 45-48 Include references to Elling et al., 2014, 2015 & 2017. Nevertheless, I don't

think you represent current knowledge on the source of OH-GDGTs fairly here. See Lipp et al., 2009; Zhu et al., 2016; Sollai et al., 2019.

Response: References were included, and we changed this sentence to "Culture studies suggest that Thaumarchaeota Group 1.1a (e.g., Nitrosopumilus maritimus) (Elling et al., 2014, 2015, 2017; Lipp and Hinrichs, 2009; Liu et al., 2012), SAGMCG-1 (e.g., Nitrosotalea devanaterra) (Elling et al., 2017), and a strain of thermophilic euryarchaeota Methanothermococcus thermolithotrophicus could synthesize OH-GDGTs (Liu et al., 2012)." Pls see our related changes in lines 47–50.

Line 167-170: As nearly half of your samples were collected in years not represented in the WOA13 V2 product, I recommend you update the manuscript with the WOA18 data that was released this summer. Furthermore, given that the studies you contrast your results to later on calculate seasonal averages from different months than those defined in the WOA product, you should use the monthly data from the WOA18 instead of the pre-defined seasonal means to strengthen the comparisons you draw in the discussion section.

Response: Thanks for comments. We updated the MS with WOA18 data. We prefer to use the pre-defined seasonal means in WOA product, because after comparing monthly mean SST in studied sites, we find that three coldest months are from Jan to Mar and warmest months are July, August and September. The definition method of 4 seasons by WOA18 already better reflect the seasonal variance of SST in this study area.

Line 177: You use the acronym "WD" yet this acronym was never defined previously in the text.

Response: Defined. Pls see line 93.

Line 205/Section 3.4: You should separate the results related to iGDGTs and those related to OH-GDGTs.

Response: Separated. Pls see.

Line 223-224: Are these relationships significant?

Response: p value was added. These relationships are very significant. Pls see related statements in lines 414–415.

Line 233-234: Why do you refer to the Chinese Marginal Sea here instead of the South China Sea?

Response: There is no local RI-OH-SST calibration for the SCS. However, the calibration proposed by Lü et al. (2015) is based on data from both East China Sea and South China Sea, which together is called the Chinese Marginal Sea.

Line 235-236: It is confusing that you refer to residuals here when the figure you reference (Fig 2) does not show the Proxy-Obs. SST residuals.

Response: In the Fig. 2, we only can see the difference between proxy-derived and observed SST. So we changed "Fig. 2 and Suppl. Table 5" to "Suppl. Table 5".

Line 239: Include references to culture studies that support this statement.

Response: We added references "(Conte et al., 1998; Prahl and Wakeham, 1987; Prahl et al., 1988; Sawada et al., 1996; Volkman et al., 1995)". Pls see lines 266–267.

Line 244-245: You really should discuss why you think your results indicate that U37K'-SST are biased towards spring temperatures as there is nothing in Figure 2 that demonstrates this. You could do a simple linear regression between U37K'-derived temperatures and seasonal SSTs and if there is a significant relationship with observation spring SSTs, then your claims are valid.

Response: Thanks for the comments. We thought that here you want to say "do a simple linear regression between U37K' index and seasonal SSTs". This method is really a common practice, but in our study region, spatial SSTs in each season varied in a very small range, with the largest in winter but still <6 °C. Together with influences

of factors other than SST on proxies, this usually leads to poor SST-proxy correlations for all seasons, albeit slightly better for winter data. So we did not use correlation as a criterion to decide seasonality. Instead, we used another common criterion, i.e., temperature residuals between calculated temperatures from established calibrations and measured seasonal SS. We used this method for consistency in the whole paper. Pls see related statements in lines 195–199.

Line 270: Either here or in the methods section you should provide more details on how you conducted this statistical analyses.

Response: We added details in method section. Pls see our related statements in lines 138–141.

Line 279: Based on your above discussion on the different sources of these three lipids and the results of the previous studies on LCD distributions in coastal environments that you cite, this supposition about an opposite relationship between C28 1,13, C30 1,13, and C32 1,15 diols and temperature is unlikely and seems unnecessary to include.

Response: This sentence was deleted.

Line 285 - 287: The statement explaining why a threshold of %C32 1,15 <20% was used from Lines 288-290 should be moved here.

Response: Sentences have been rearranged. Pls see lines 313–315 and 317–320.

Line 305: Replace "methane-related" with something more accurate such as "...iGDGTs from archaea involved in methane cycling..."

Response: Replaced. Pls see lines 341–342.

Line 306: Please indicate what "substantially elevated" values for the [2]/[Cren] and [0]/Cren indices are as you do for the MI.

Response: "substantially elevated" values for the [2]/[Cren] is >0.2, and [0]/[Cren] indices is deleted from here. Pls see line 340.

Lines 309-312: Another paper you cite elsewhere in the text, Zhou et al. (2014), concluded that brGDGTs in the PRE are also likely derived from in situ production in the river rather than solely originating from erosion of catchment soils.

Response: The ability of the BIT index to indicate soil input in this region has recently been discounted by finding that branched GDGTs may be aquatically in-situ produced (Zhou et al., 2014). Nevertheless, considering that the sample PRE-A8 is located at the upper river mouth, together with the highest %C32 1,15 values as discussed above, we believe iGDGTs may be impacted to some extent by terrestrial input. Pls see related statements in lines 345–3448.

Line 321-322: The difference in iGDGT distributions between 'deep' and 'shallow' Thaumarchaeota eco-types is due to the use of different enzymes for iGDGT synthesis (cf. Kim et al., 2016 doi: 10.1016/j.gca.2015.09.010; Villanueva et al., 2014 doi: 10.1111/1462-2920.12508).

Response: Changed. Pls see lines 360–361.

Line 339-343: In Kim et al.'s 2008 paper, the authors note that the difference in their TEX86 calibration for core top sediments from depths <200 m relative to the calibration for the entire data set is negligible, likely because contribution of iGDGTs from deep dwelling archaea to the sediment floor is minimal relative to the contributions from shallow-dwelling Thaumarchaeota, a point that has been highlighted in many other studies. As such, I don't believe this is the cause of the mismatch between the TEX86H-reconstructed temperatures and observations you report in your manuscript.

Response: We note that the influence of water depth on the TEX86 proxy have not reached an agreement. The contribution of iGDGTs from deep dwelling archaea to the sediment floor has been estimated to be >50% in the deep seas (e.g., Kim et al., 2016; Jia et al., 2017), although the abundance of Thaumarchaeota and GDGTs have

been found maximum at the lower euphotic zone. We thought the contribution of deep dwelling archaea might be a background for sedimentary GDGTs in a specific site, where TEX86 could be mainly controlled by the variable shallow-water iGDGTs. Nevertheless, when considering spatial distributions, the contribution of the deep dwelling archaea could change some extent, which may be a cause of the significant TEX86-SST scatters. Of course, these are beyond this paper, and we did not say more about that in the paper. Kim, J.-H., L. Villanueva, C. Zell, and J. S. Sinninghe Damsté (2016), Biological source and provenance of deep-water derived isoprenoid tetraether lipids along the Portuguese continental margin, Geochim. Cosmochim. Acta, 172, 177–204. Jia, G., X. Wang, W. Guo, and L. Dong (2017), Seasonal distribution of archaeal lipids in surface water and its constraint on their sources and the TEX86 temperature proxy in sediments of the South China Sea. J. Geophys. Res. Biogeosci., 122, 592–606.

Line 390: In Line 315 you draw the opposite conclusion that your samples are not appreciably impacted by TEX86H?

Response: We suspect "TEX86H" could be "by soil input". Our opinion is that judged from the BIT value, the sample of PRE-A8 is influenced by soil input (Lines 345–347). However, [Cren'] data could also suggest the predominance of Euryarchaeota (Lines 354–357) for the sample. We think this is not controversy as the two factors may co-occur at this site.

―――――――――――――――――――――

---

## Author Comment (AC2) · 14 Dec 2019

Dear reviewer, Thank you for your constructive comments. We have addressed your comments and make changes accordingly. Please find related contents in the document file with changes marked.

Section 2.1: How as the sedimentation rate determined (1–2 mm/yr, stated in line 170)? The sediment accumulation rate is likely to vary across the offshore transect. Therefore, assuming that all core-top samples represent the mean conditions of a 7-year interval (from 2005–2012) is probably not appropriate. This assumption also does not account for bioturbation, which almost certainly has caused some mixing of material

from the past few decades into the upper few centimeters. Making sure that the core-tops in this study are "calibrated" to observational temperatures from the appropriate time interval is especially important given the large SST trends observed over the past decade in the SCS (e.g., Yu, Y., Zhang, HR., Jin, J. et al. Acta Oceanol. Sin. (2019) 38: 106. https://doi.org/10.1007/s13131-019-1416-4).

Response: Thanks for the comments. The sedimentation rate here is unknown. In the revision, we updated SSTs data from WOA18 (from 2005–2017), a wider time interval could better cover sampling time. Although SST have increased over the past decade in the SCS, but it has less influence to the average SST data within the time interval (e.g., WOA13 vs. WOA18, the average difference is less than 0.5 °C, which is lower than the calibrations error for each proxies).

Uk37-derived temperatures: A number of studies have pointed out the non-linearity of the U37K'-Temperature relationship at SSTs >24–26°C (e.g., Sonzogni et al., 1997, Conte et al., 2006, Tierney and Tingley, 2018). Since SSTs are >24°C for most of the year at these SCS core sites, it would be worthwhile to calibrate the U37K' data in this study using BAYSPLINE (Tierney and Tingley, 2018), which accounts for the attenuation of the U37K' signal at higher SSTs.

Response: We have checked the difference between linear and nonlinear calibrations, especially the BAYSPLINE (Tierney and Tingley, 2018) SST estimate. Indeed, BAYSPLINE SST estimate yields slightly higher temperature values by ∼0.5 °C in average than the linear calibration. But this is not against our conclusion of spring bias of alkenone temperature but reinforces it. Pls see related statements in lines 218–220 and 275–277.

Supplemental figure 1. If the U37K'-derived SST ends up being significantly different using alternative U37K'-SST equations, it will have implications for the inferred seasonality of the U37K' signal and needs to be addressed more thoroughly in the main body of the paper.

Response: In the revision, we moved the supplementary figure 1 to Fig. 3, and all derived U37K'-SSTs exhibited similar values that differed from the calibration of Conte et al. (2006) by <0.5 °C (0.2 °C average).

Lines 60–79: There is a substantial body of literature discussing sources of uncertainty and biases in these four biomarkers (especially TEX86 and Uk37) from sediment traps, surface sediments and culture studies. This section should be expanded to include some discussion of those factors. Just a few examples: lateral advection of sediments, light limitation, diagenesis (e.g., preferential degradation of C37:3 in sediments), sensitivity to redox conditions, etc.

Response: In the revision, we add "Nevertheless, environmental and physical parameters may also bias these proxies, including: (1) lateral advection (Benthien and Müller, 2000; Kim et al., 2009); (2) different resistance to degradation (Goni et al., 2001; Kim et al., 2009); (3) nutrient stress and light limitation (Hurley et al., 2016; Park et al., 2019; Prahl et al., 2003; Versteegh et al., 2001)." Pls see related changes in lines 71–74.

Line 167: Why not use the latest WOA18 data?

Response: We updated our data with the latest WOA18.

Line 181–182: Looking at the local hydrographic data, the depth of the mixed layer appears to vary seasonally, with a much deeper winter mixed layer. This should be considered when discussing the potential seasonal and/or depth distribution of the biomarker source organisms.

Response: The use of the water column structure data in phytoplankton ecology studies is really a common practice. However, our study sites are mostly shallow with water depths of most of them <50 m, where water stratification does not extensively occur and WOA data fail to provide high-resolution data. So we didn't rely our interpretation on those information.

Overall, there needs to be a much more thorough treatment of uncertainty in the

manuscript. There are no error bars or uncertainties shown in any of the figures, nor are they discussed in the results or the supplementary data table. Uncertainty in the various transfer functions used to convert each of these indices to temperature needs to be considered. Analytical uncertainty could be addressed via replicate measurements of samples or standards. The analytical error and calibration error should be propagated and reported when converting proxy index to SST.

Response: In the revision, both calibration and analytical errors are considered and described in the Method section. Our analytical errors for different proxies are much lower than calibration errors. Pls see related contents in lines 114, 116–118, 132, 138, 166, 170–172, 181, 185–187.

Lines 198–204: In order to maintain the organizational flow of the paper, this section belongs in the Discussion. Also, instead of making the argument that you omit these 6 samples because of their large SST residuals, it makes more sense to omit them because the river input index (%C32 1,15) values are 4x higher than those of the other 19 samples.

Response: Rephased. Pls see related contents in lines 317–320.

Section 4.2.2: Is there any correlation between the BIT and diol river input index in this sample set?

Response: BIT exhibited a linear relation with diol river input index ($R^2 = 0.66$, $p < 0.001$). We added it in line 320.

TEX86: There is a large body of literature on the TEX86 proxy that is overlooked in this manuscript. Marine Thaumatchaeota are living throughout the water column in many locations, and it's likely that the TEX86 is integrating the entire water column in these shallow (<200 m) sites. As with the Uk37-SST equations, I would suggest including BAYSPAR-derived SSTs in the TEX86-SST analysis (Tierney and Tingley, 2015).

Response: We re-evaluated our data using BAYSPAR according to the comment, and

did not find substantial differences. Comparatively, the local "shallow-water" calibration yielded smallest temperature residuals. Pls see related discussion in lines 392–393 and 398–402.

Section 4.3.1: The Ring Index (RI), as defined by Zhang et al., 2015, could easily be calculated from the GDGT data used to calculate TEX86 in this study. This is another tool (in addition to the MI, [2]/[cren], and [0]/[cren]) that could be used to screen for non-thermal influences on iGDGT distribution in this sample set.

Response: We admitted that we need to be thoughtful. In the revision, we added these related contents. Pls see our discussion in lines 368–377.

Figures: The uncalibrated index values (uk37', LDI, RI-OH and TEX86) are not reported in your figures. I think there should be at least a table that shows the primary data from each of the core-top sites.

Response: In the revision, we added related contents in Table 1. Pls see.

Figure 2: This figure is not a very effective way to present these data. I would consider presenting SST maps to show the mean annual, winter and summer SST distribution in the study area (showing spring and autumn is unnecessary in my opinion). Perhaps create separate panels for each of the SST indices. If the authors decide to keep this figure, the lines connecting WOA data points need to be removed, they are distracting. The use of the same colors for the WOA SST data and the proxy-SST data is confusing.

Response: In the revision, this figure was split to a 4-panel figure with 1-panel per proxy. Although the lines connecting WOA data still exist, but they have no impacts on showing the differences between WOA SST and proxy SST.

Figure 4: As with Figure 2, panels a and b are not a very informative. Dividing the samples into "inshore and offshore" as is done in Figure 5 and presenting the relative abundances and ratios/indices as bar graphs would be more effective. Again, if the authors chose to keep these panels, refrain from connecting data points with lines.

Response: In the revision, we changed related figures. Pls see Fig. 5.

Figures 3 & 5: The inshore versus offshore comparison of the fractional abundances of diols in figure 5 illustrates the same point as the maps in Figure 3. Therefore, I think Figure 3 is unnecessary. It could be moved to the supplement or removed altogether without losing any information. Figure 6: I would suggest adding panel d from figure 3 as a second panel in Figure 6 to better illustrate the elevated influence of terrestrially sourced diols in the PRE.

Response: Yes, there are some repetition between Fig. 3 and Fig. 5, but we want to keep this map, because it not only emphasizes the similar spatial variation of C28 and C30 1,13-diols and C32 1,15-diols, but also exhibits the "unusual" data points. Fig. 5 emphasizes the comparison of LCDs composition between SPM and surface sediments in this study. In the revision, we combined Fig. 3, 5 and 6 in the new Fig. 4. Pls see.

Figure 7: There is no reason to plot the residuals of 3 different calibrations versus the BIT index. If the purpose of the figure is to illustrate that there is no systematic relationship between BIT and SST residuals, then you need only illustrate this using one of the calibrated data sets. If the purpose of the figure is to illustrate the calibration equation that results in the smallest residuals, I would suggest a simpler way of showing the distribution of the data (e.g., box and whisker plots).

Response: The purpose of this figure is to show that residuals from local calibration is smallest compared to global and Bayspar calibrations. Following your suggestion, we changed it to box and whisker plots (Fig. 6b).

Figures 7b & 9: I don't think it's terribly informative to regress any of these indices over a <2 °C temperature gradient (as in the case of summer and autumn SST gradients), however, if you are going to make this plot, why not make a 4-panel figure that does the same for all 4 indices?

[Figure]

Response: Yes, linear regression is not an appropriate method applied to such narrow temperature intervals here. In the revision, we deleted this plot and related discussion.

Figure 8: The source of the data from a "previous study" needs to be cited in the caption.

Response: In the revision, Fig. 8 was moved to Fig. 7, and the source of data was added. Pls see the caption of Fig. 7.

Figure 8: In the caption, the authors need to clearly state what "annual residuals" are. Also, "fitting lines" is not a mathematical term. Are these ordinary least squares regression lines? Something else?

Response: The explanation of "residuals" could see Eq. (14). We added related statements in the caption. Pls see the caption of Fig. 7.

Minor Comments: Lines 49–50: This sentence is awkward to read. I would suggest changing to something like "Due to the distinctive ecology of their source organisms (e.g., depth habitat and seasonal preference), coeval temperature records from each of these proxies may differ substantially"

Response: Changed. Pls see lines 51–52.

Line 51: Remove the "however" from this sentence.

Response: Removed.

Lines 57–59: Remove "while" from the beginning of the sentence. These two sentences seem to contradict each other. Also, what is the southeast Australian Ocean?

Response: "while" is removed. We think that these two sentences does not contradict each other. We change "southeast Australian Ocean" to "Australian southern and eastern coasts". Pls see in line 60.

Line 60: This sentence doesn't make sense: "The accuracy of organic thermometers

is also prone to be impaired by a low specificity of related biomarkers."

Response: We changed "The accuracy of organic thermometers is also prone to be impaired by a low specificity of related biomarkers." to "The accuracy of organic thermometers is also interfered by the diverse origins of related biomarkers."

Line 83: Should be northeasterly and southwesterly winds

Response: Changed. Pls see line 88.

Lines 171–172: I believe WOA13 defines the summer and autumn as Jul–Sept and Oct–Dec, respectively (not Jul–Aug and Sept–Dec as stated here).

Response: Changed. Pls see line 194.

Line 199: Instead of stating "three samples with low LDI values", it would be more descriptive to state, "three samples with LDI values lower than predicted from local SST".

Response: This sentence was deleted in the revision.

Line 244: Explain what is meant by "complex sedimentation processes". If the authors are talking about lateral transport of the fine sediment fraction, or diagenetic alteration of the signal, this could be expanded on significantly here.

Response: "complex sedimentation processes" means lateral advection and resuspension processes. Due to a narrow SST intervals, the impact of lateral advection may be minor. For examining the diagenetic alteration, it is better to use SPM from different water depths or downcore sediments. It is beyond our paper, because our samples are core-top sediments.

Lines 257–258: This sentence is confusing and should be rewritten for clarity.

Response: This sentence was deleted. Because if we take the calibration error and analytical errors into consideration, the slightly lower UK37'-SST observed in the river

mouth is possibly due to the above errors.

Line 279: It is unclear what is meant by "an opposite response to ambient temperature of C30 1,15-diol to C28 and C30 1,13-diols".

Response: Deleted.

Line 305: Change "methane-related" to methanotophic archaea

Response: In the revision, we changed "methane-related" to "archaea involved in methane cycling". Pls see lines 341–342.

―――――――――――――――――――

---

## Referee Report (RR2)

I am very pleased that this version of the manuscript is infinitely easier to read and absorb than the version I reviewed before. The authors should be credited with a thorough revision and response to earlier reviews. The paper now reads like a nice introduction to the use of the 4 organic proxies evaluated here, and has some important take away points.

As the authors acknowledge, the small annual temperature range in the study area limits some of the resolution on what they can say about the different temperature proxies. As they say, they also cannot distinguish the importance of annual production relative to monsoon transition period, as that time of the year has an SST identical to mean annual. However, the coastal SCS location does have some virtues: the region has a large seasonal temperature range and the sediments also receive significant terrestrial/riverine inputs, allowing the authros to assess the confounding influence of non-marine inputs.

Coastal sediments do have a disadvantage in the complexity of organic matrices. In my experience, complete separation of alkenones from co-eluting peaks can be very challenging. The authors have done standard purifications, but the use of a of 6oC/minute GC temperature during alkenone elution time is not optimal to resolve interferences. I would like to see more attention in the methods section to how confident the authors can be that they have uniquely identified the critical compounds for each proxy. Some representative (i.e. not the "best") chromatograms as supplemental material would be helpful for a critical reader.

The evaluation of the different proxies seems quite reasonable. The authors do have to exclude a fair number of outliers, but, since the raw data are included, and the criteria explicitly discussed, I think the data culling is appropriate.

I would urge the authors to be more critical of the utility of the TEX index:

"The better agreement of $TEX_{86}$ temperature estimates with EAWM SSTs suggests that conditions during the EAWM period may be favourable for the bloom of the autotrophic ammonia oxidizing Thaumarchaeota, the activity of which is enhanced at low light availability and high ammonia concentrations "

In fact, the TEX index-derived SST departs badly from ANY seasonal SST and also shows very wide scatter in comparison to the other proxies

And later:

Overall, we observed good agreement between measured annual mean or seasonal SSTs and temperature estimates based on 510 the four proxies discussed here, after excluding samples with obvious signs of terrigenous supply of the respective lipids.

This belies the very bad departure of the TEX estimates from the other proxies, and also the very large scatter in inferred temperatures- I'd say that TEX is distinctively problematic in this setting.

I would welcome the authors' response to my remaining critical comments but I would urge publication following minor revision.

---

## Author Response (AR2)

1. I applaud the authors for producing a comprehensive data set of indices biomarkers related to ocean temperature from modern sediments. Their focus on potential differences in the interpretations of the proxies as a result of seasonal differences in ecology is reasonable. The issue is whether the data set has the ability to resolve the question posed. A major obstacle is that all temperature calibrations are empirical. This means that anomalies in inferred temperature must be made PRESUPPOSING that the calibration to temperature is accurate, and then finding the best fit of the proxy to local (seasonal) temperature. I don't have a smart way out of this dilemma, but it poses a significant signal: noise issue: at what point does a proxy deviation from an expected relationship become large enough to be meaningful? A big caveat for this study region is recognized in lines 203-204: "Note that spring, autumn and annual temperatures are very close to each other. " This problem severely limits the seasonal interpretations proposed here. I was a bit surprised that the authors did not consult satellite estimates of seasonal chlorophyll patterns. This might have buttressed some of the claims made for e.g. the late spring importance of riverine runoff on production. In the end, I was left unconvinced that the data sets had the resolving power to support the proposed seasonal biases in biomarker production. Some examples: a spring bias for alkenones claimed, but then later (page 25, ) that "alkenone production is biased to warm season". In this regard, using a definition of May to June as "spring" is hard to follow- there can't be much discriminatory power in using SST by this definition. Furthermore, as the authors acknowledge, the difference between spring and mean annual SST is minute.

**Response:** Thanks for the critical comments. We acknowledge that in the warm/hot (sub)tropical region where our study was conducted, it is not feasible to divide the year into 4 seasons. In fact, the climate in our study area is characterized mainly by two contrasting seasons: the warm and wet season dominated by the southwesterly East Asian summer monsoon (EASM) from May to September and the cold and dry season dominated by the northeasterly East Asian winter monsoon (EASM) from December to February. The so-called spring and autumn are relatively short and transitional. Accordingly, in the revised version of our manuscript, only two seasons, EASM season and EAWM season, are defined, which is different from the summer (July-September) and winter (January-March) in the previous version. The SST differences between EASM and EAWM seasons are large enough for us to see whether they were reflected preferentially by a proxy-derived temperature. Of course, we still cannot differentiate the transitional periods, i.e., October-November and March-April, from the annual simply by the SST data; however, due to their transient nature, we do not treat them as primary but secondary seasons. Nevertheless, we take them into account at some places in our discussion, e.g., in lines 320-327; 376-378.

2. Likewise, the difficulty in interpreting the meaning of the TEX86 values becomes apparent (p. 28-29) and it's hard to support a seasonal mechanism.

**Response:** GDGTs, as well as the $TEX_{86}$ proxy, have been extensively studied in the last decade. As the archaea origin, which synthesize GDGTs, are ubiquitous in various environments, there are numerous confounding factors on the ability of

TEX$_{86}$ to reflect SST, including the water depth of the Thaumarchaeota's preferential habitat, terrestrial soil input, contributions of planktonic and methane-related Euryarchaeota, etc. Therefore, several indices constraining the application of TEX$_{86}$ have been developed, which can be used to examine whether TEX$_{86}$ in a given sample is suitable to estimate SST. We followed this conventional procedure by considering indices such as the MI, BIT, [2]/[Cren] ratio, [2]/[3] ratio, [Cren'] and RI to assess the impacts of non-thermal factors on iGDGTs composition. Our results show that except two samples, these indices qualify all our iGDGTs data as being likely to reflect SSTs. Whichever calibrations were used, including the widely used global calibration, the newly proposed BAYSPAR, and a local winter calibration, the estimated temperatures were always lower or close to winter monsoon season SSTs. So we believe that our association of TEX$_{86}$ temperature with winter monsoon is robust.

3. For the RI-OH calibration, residuals range from 1.5 to 1.9 oC for the 4 seasons- simply picking the season of lowest residual does not make a compelling statistical argument.

**Response:** Thanks for the comments. A major revision has been made on the discussion of OH-GDGTs. Firstly, we now use a summer calibration of RI-OH from Lü et al. (2015), because RI-OH correlated best with the summer SST in that work (Lines 205-210). Then we showed that the standard error of the average residual yielded from the summer calibration is much lower than from annual and winter calibrations (section 4.4.2), indicating that the summer calibration can provide the most precise estimates. Accordingly, we speculate that the source organisms proliferate mainly during the EASM season.

4. Abstract: The spring bloom is more typically triggered by winter mixing followed by increasing sunlight and stratification, not riverine input of nutrients. Is there evidence to support the importance of riverine input to these sites? The claim that eustigmatophytes would be "insensitive to nutrient inputs" is hard to believe since these are photoautotrophs. Controls on GDGT production left very vague: "relatively high nutrients levels, low light and high concentrations of SPM". Of these, nutrients in the usual sense does not apply to nonphotosynthesizing archea (some of which are believed to be ammonium oxidizers as mentioned in the Introduction, others chemautotrophs) nor do light conditions seem relevant. Lines 68,69: "Recently, the LDI proxy was found to be limited by the input of 1,13 and 1,15-diols from freshwater eustigmatophyte algae, especially in the coastal seas" The meaning of "limited" is not clear. Presumably the sense is that the LDI proxy departs from an expected calibration because of the input of the 1,13 and 1,15 diols?

**Response:** Thanks for these comments. We rephrased the abstract, and the sentences criticized by the reviewer have now been removed.

5. Section 2
Since inferences of seasonal production rely on the deviation of the measured indices relative to temperature from reference calibrations, and those deviations may only be a few degrees, the quality of the analyses

becomes very important. The quality of the analytical determinations may be critical, especially in the case of shallow water sediments, which typically contain a wide variety of organic compounds, many of which may interfere analytically with the compounds of interest. Insights (and examples?) of the degree to which lipid extraction and separation removed interferences would be very helpful. In this regard, the method used for alkenone determination employs a rather fast temperature ramp and would not be ideal to resolve co-eluting compounds, unless the fraction   nalysed is very pure. The uncertainty reported for alkenone Uk'37 determination comes from a reference standard- it would be very valuable to know the reproducibility of the SCS sediments, with full replication of extraction, separation, and GC analysis. The same caveat applies to analytical uncertainties in the GDGT and RI-OH method.

**Response:** Thanks for this comment; we agree that the quality of the analyses is a crucial point. However, we believe that our laboratory protocol yields very high quality measurements, and we take care to clean our fractions in the best possible. As described in the manuscript, saponification was used to eliminate the interferences from the sedimentary wax esters, which has been suggested as a crucial clean-up procedure for alkenone analysis (Villanueva et al., 1997) (Lines 113-114). A full replication of extraction, separation, and GC analysis was unfortunately not done during our experiments. However, based on a typical chromatogram (for sample E503) illustrated below, we are confident that interferences have been satisfactorily removed and $C_{37:2}$ and $C_{37:3}$ alkenones are completely separated. As to methods for long-chain diols, iGDGTs and OH-GDGTs, they were instrumentally   nalysed using GC-MS or LC-MS in selective ion monitoring (SIM) mode, the reproducibility of which does not rely heavily on removal of co-eluting substances but rather on the sensitivity of the instrument and amount injected. The sample pre-treatment we used were also used by some of the participants in the interlaboratory comparison of TEX86 analytical methods, where extraction procedures were not found to exert significant and systematic effects on TEX86 results (Schouten et al., 2013b, doi:10.1002/2013GC004904).

[Figure]

6. I am not an expert in the LDI analysis. However, the sentence beginning at line 138 concerning the use of a Pearson correlation coefficient is very unclear (what is being correlated to what, and why is this a measure of

uncertainty?)

**Response:** In the revision, details about Pearson correlation analysis were added, please see lines 261-264.

7. Climatological temperatures: Typically, coastal SST shows a lot of heterogeneity. Is the resolution of the WOAA gridded data set sufficient to define the climatology at the resolution of the core sampling?

**Response:** The grid resolution of 0.25° in the database is sufficient to define the climatology of the study region, as the distances between most (n = 19 of 23) sampling sites are >0.25° (Fig. 1). Please see Fig. 1.

8. Section 3
Given that deviations relative to expected values are key, much attention needs to be devoted to the uncertainties inherent in estimating deviations/residuals. Although the authors have responded about their non-use of the Bayspline, their use of the Conte 2006 calibration would not be the choice of most in the alkenone field (the standard reference would be to Muller et al., ). To side-step a debate on which calibration is "best", a more rigorous effort is needed. The authors do not report the mean and standard deviation of their Uk'37 values. This would let the reader assess better whether the data support any attribution to a season of production. My guess is that the alkenone data agree to within error to spring, autumn, or annual production. Similarly, reporting the mean and standard deviation of LDI estimates (screened to remove terrestrial inputs) would be very useful. Given the low apparent temperature derived from the coastal sites using the TEXH86 values, how do the authors assess the importance of contamination from terrestrial inputs as opposed to a marine production signal? The discussion of the relation of TEX86 together with the Ring Index is confusing for a nonspecialist. The claim at first appears to be that the TEX index should not be used to estimate temperature, but then the claim is contradicted. I appreciate that resolving proxy uncertainties isn't easy, but this section isn't very satisfying.

**Response:** Thanks for the comments. In the revision, (1) we added analysis and calibration errors to all proxy-derived temperatures (Fig. 2a, 2b, 2c and 2d), and also included them in the discussion; (2) BAYSPLINE and the calibration from Muller et al. (1998) are examined for $U^{K'}_{37}$; (3) The influence of terrestrial input on iGDGT compositions has been carefully discussed in section 4.3.1. (4) As for the $TEX_{86}$-RI relationship, theoretically, if planktonic archaea are the dominant GDGT producer, the RI values involving all iGDGTs compounds reflect a response to growth temperatures similar to $TEX_{86}$, leading to a positive correlation between the two indices. Accordingly, Zhang et al. (2016) proposed a global $TEX_{86}$-RI relationship, which can be used as a criterion to see whether the $TEX_{86}$ value of a given samples is influenced by non-thermal factors. We found most of our sediment data showing a good correlation between $TEX_{86}$ and RI (Fig. 6a); however, they plot outside of the 95% prediction band of the global $TEX_{86}$-RI relationship (Fig. 6b) but are within the 95 % prediction band of a "shallow-water" $TEX_{86}$-RI relationship (Fig. 6b). We suggest that this "shallow-water" $TEX_{86}$-RI relationship that is different than that of the global core-top data set is a robust feature. Our study sites receive predominantly shallow Thaumarchaeota input as demonstrated above, and the shallow Thaumarchaeota likely responds to ambient temperature

differently from the deep dwelling communities (Jia et al., 2017; Kim et al., 2015, 2016; Taylor et al., 2013; Villanueva et al., 2015; Zhu et al., 2016). Similarly, the $TEX_{86}$ and RI values from an incubation study of marine Thaumarchaeota (Schouten et al., 2007) are correlated but lie outside of the 95 % prediction band of the global relationship, likely due to differences in the archaeal community between the incubation experiment and natural marine settings (Zhang et al., 2016). Together, this indicates that TEX86 is suitable for temperature estimation in our study area and $TEX_{86}$ in most of our sediments likely indicate regional seawater temperatures.

**Reply to referee 2**

Suggestions for revision or reasons for rejection (will be published if the paper is accepted for final publication)

While I do think the paper provides some interesting new data, particularly in regard to the OH-GDGTs and long chain diols, the manuscript still has several issues that I feel make the article unsuitable for publication in this journal in its current form. First, the english of the manuscript still needs to be carefully reviewed and all spelling errors need to be corrected (e.g. line 18 "Specially"). Furthermore, I noticed an error in one of the supplemental tables (S. Table 4, mean winter absolute offset value) and that the authors did not report BAYSPAR or BAYSPLINE-calibrated temperatures in their tables even though the data was discussed in the text. Aside from such technical errors, I feel the authors did not adequately address reviewer comments in regard to:

1. the incorporation of uncertainty into figures and the discussion

2. errors that may have arisen due to their chosen method of lipid extraction,

3. proxy calibration choice/justification when both reviewers noted that the current literature does not support their use of the linear UK'37-SST calibration nor TEX86H,

4. expansion of the discussion to include considerations of processes other than seasonality and terrigenous input on sedimentary lipid distributions in the South China Sea.

**Response:** We carefully revised our manuscript this time, including English corrections, missing data replenishment, incorporation of uncertainties into figures and the discussion, analytical errors, and taking the non-linear calibrations of $U^{K'}_{37}$ and $TEX_{86}$ into account. We appreciate the reviewers to examine our amendments.

As to the processes other than seasonality and terrigenous input on sedimentary lipid distributions, we speculate the reviewer referred to sediment resuspension and lateral transport raised by previous comments. We are not able to provide a comprehensive discussion on this process due to lack of any evidence. However, we observed good agreement 
[revised manuscript text omitted]

925

---

## Author Response (AR3)

**Editor's comments**

Your contribution has now been re-reviewed by one referee. Based on their report and my own reading of both the manuscript as well as all the referee reports of prior iterations, your manuscript should be acceptable for publication following minor changes. In addition to the minor edits specified in the referee report, I would encourage you to acknowledge in more detail prior work on UK37' in the South China Sea, i.e., the paper by Pelejero & Grimalt (1997) already discusses many of the issues you raise in your contribution, including seasonality.

**Response:** Thanks for editor's comments. We cite the work of Pelejero and Grimalt (1997) in the revised manuscript, please see lines 51-52, 316-319.

**referee #3**

1. I am very pleased that this version of the manuscript is infinitely easier to read and absorb than the version I reviewed before. The authors should be credited with a thorough revision and response to earlier reviews. The paper now reads like a nice introduction to the use of the 4 organic proxies evaluated here, and has some important take away points. As the authors acknowledge, the small annual temperature range in the study area limits some of the resolution on what they can say about the different temperature proxies. As they say, they also cannot distinguish the importance of annual production relative to monsoon transition period, as that time of the year has an SST identical to mean annual. However, the coastal SCS location does have some virtues: the region has a large seasonal temperature range and the sediments also receive significant terrestrial/riverine inputs, allowing the authors to assess the confounding influence of non-marine inputs.

Coastal sediments do have a disadvantage in the complexity of organic matrices. In my experience, complete separation of alkenones from co-eluting peaks can be very challenging. The authors have done standard purifications, but the use of a of 6° C/minute GC temperature during alkenone elution time is not optimal to resolve interferences. I would like to see more attention in the methods section to how confident the authors can be that they have uniquely identified the critical compounds for each proxy. Some representative (i.e. not the "best") chromatograms as supplemental material would be helpful for a critical reader.

**Response:** Thanks for the critical suggestion. We are confident that compounds for each proxy have been clearly separated in our experiments, as shown by chromatograms in Fig. S1.

The evaluation of the different proxies seems quite reasonable. The authors do have to exclude a fair number of outliers, but, since the raw data are included, and the criteria explicitly discussed, I think the data culling is appropriate. I would urge the authors to be more critical of the utility of the TEX index: "The better agreement of TEX86 temperature estimates with EAWM SSTs suggests that conditions during the EAWM period may be favourable for the bloom of the autotrophic ammonia oxidizing Thaumarchaeota, the activity of which is enhanced at low light availability and high ammonia concentrations". In fact, the TEX index-derived SST departs badly from ANY seasonal SST and also shows very wide

scatter in comparison to the other proxies. And later: "Overall, we observed good agreement between measured annual mean or seasonal SSTs and temperature estimates based on the four proxies discussed here, after excluding samples with obvious signs of terrigenous supply of the respective lipids." This belies the very bad departure of the TEX estimates from the other proxies, and also the very large scatter in inferred temperatures- I'd say that TEX is distinctively problematic in this setting. I would welcome the authors' response to my remaining critical comments but I would urge publication following minor revision.

**Response:** Thanks for the critical comments. We noticed that many TEX estimates were even lower than the SSTs in cold months that was not properly treated in the last version. In the updated version, we refer to prior works of Wang et al. (2015; 2017), in which the lower TEX estimates were attributed to the contribution of iGDGTs 1 to 4 from Marine Group II *Euryarchaeota*. Nonetheless, more works are needed to reach a final conclusion. As the relatively poor performance of TEX estimates in the coastal SCS, we carefully rephase the wording in the part of TEX. Please see in lines 25, 458–463, 516–519, and 546.

[revised manuscript text omitted]